# Biomedical knowledge graph learning for drug repurposing by extending guilt-by-association to multiple layers

Dongmin Bang[1,2], Sangsoo Lim[3], Sangseon Lee[4] & Sun Kim [1,2,5,6] ✉

Computational drug repurposing aims to identify new indications for existing drugs by utilizing high-throughput data, often in the form of biomedical knowledge graphs. However, learning on biomedical knowledge graphs can be challenging due to the dominance of genes and a small number of drug and disease entities, resulting in less effective representations. To overcome this challenge, we propose a "semantic multi-layer guilt-by-association" approach that leverages the principle of guilt-by-association - "similar genes share similar functions", at the drug-gene-disease level. Using this approach, our model DREAMwalk: Drug Repurposing through Exploring Associations using Multi-layer random walk uses our semantic information-guided random walk to generate drug and disease-populated node sequences, allowing for effective mapping of both drugs and diseases in a unified embedding space. Compared to state-of-the-art link prediction models, our approach improves drug-disease association prediction accuracy by up to 16.8%. Moreover, exploration of the embedding space reveals a well-aligned harmony between biological and semantic contexts. We demonstrate the effectiveness of our approach through repurposing case studies for breast carcinoma and Alzheimer's disease, highlighting the potential of multi-layer guilt-by-association perspective for drug repurposing on biomedical knowledge graphs.

Novel drug development process in the modern era is costly, both in terms of resources and time. Drug repurposing utilizes already-approved drugs to treat diseases, and it is increasingly becoming an attractive alternative for treatment-lacking conditions. The benefits of using existing drugs lie in the lower risk of toxicity-related clinical failure, along with lower development costs and shorter approval timelines[1].

Accumulating bioassays and screening results have led to better-than-ever understanding of drugs and diseases at the molecular level. Computational drug repurposing has gained attention owing to its rapidness and ability to utilize high-throughput data[2], especially with

the rise of the pandemic era[3]. Throughout the COVID-19 pandemic, a number of computational methodologies have been successful in predicting the use of existing drugs for COVID-19 patients. A notable example is an expert-curated network analysis that discovered baricitinib[4], which is now approved by the US Food and Drug Administration (FDA) in combination with remdesivir[5]. As another example, a transcriptome, proteome, and human interactome-integrative network approach along with population-based study identified melatonin as a potential prevention and treatment for COVID-19[6]. As the number of drug repurposing cases grew, so did the interest in a systematic data-driven, instead of hypothesis-driven,

[1]Interdisciplinary Program in Bioinformatics, Seoul National University, Seoul 08826, Republic of Korea. [2]AIGENDRUG Co., Ltd., Seoul 08826, Republic of Korea. [3]School of Artificial Intelligence Convergence, Dongguk University, Seoul 04620, Republic of Korea. [4]Institute of Computer Technology, Seoul National University, Seoul 08826, Republic of Korea. [5]Department of Computer Science and Engineering, Seoul National University, Seoul 08826, Republic of Korea. [6]Interdisciplinary Program in Artificial Intelligence, Seoul National University, Seoul 08826, Republic of Korea. ✉e-mail: sunkim.bioinfo@snu.ac.kr

screen of all known drugs by fully incorporating the large bioassay datasets[7].

Many models have attempted to connect drugs to candidate disease by constructing drug-disease bipartite similarity networks[8–11]. For example, MVGCN[11] constructed a multi-view drug-drug and disease-disease similarity network for drug-disease association (DDA) prediction. However, the limitation of these methods is that they do not fully consider the biological mode of action (MoA) of drugs and their relationship with disease. A more convincing and widely-used method is clarifying biological mechanisms with relevant genes. This method has been well applied in the aforementioned cases of baricitinib and melatonin against COVID-19, where the target genes of the disease have already been intensively identified. However, this is not the case in general, where a drug's MoA needs to be inferred and this inferred MoA needs to be connected to disease. Hence, a single computational framework that connects through all three layers of drug, gene, and disease-integrated knowledge graph is required.

Several studies have been proposed to leverage the integrated drug-gene-disease knowledge graph, or biomedical knowledge graph (biomedKG) for systemic DDA prediction and drug repurposing[4, 12–15]. Himmelstein et al.[13] performed meta-path based network mining on a constructed a heterogeneous network, named HetioNet, for drug repurposing. Also, Ruiz et al[12]. analyzed the network diffusion profile of drugs on their constructed Multi-scale Interactome (MSI) network and revealed that integrating gene ontology (GO) annotations on biomedical network improved both DDA prediction performance and interpretability. A Graph Convolutional Network (GCN)-based drug repurposing model, biFusion[16], reported performance enhancement when the PPI network was integrated into a drug-disease bipartite network. Lastly, a recently proposed model iDPath[17] adopted a deep learning framework to connect drugs and diseases through a multi-layer knowledge graph for drug repurposing. iDPath identified critical paths that match drugs' MoA, implying that connection of drug and disease through the MoA-relevant path is critical for accurate prediction of DDAs.

The main research issue is that learning drug-disease association with the PPI-based gene-gene network (PPI network) brings forth technical challenges. The main hurdle is that the PPI network is much larger and denser than drug-gene and disease-gene networks. Statistics of several biomedical heterogeneous networks show that the gene-gene network covers over 90% of nodes and edges owing to its large number of entities and high degree (Supplementary Fig. 1). Connecting two sparse networks through a large and dense network is difficult, and network representation learning frameworks of other domains suffer from bias towards the PPI network. Empirical analysis of drug-gene-disease knowledge graphs showed that random walk and network propagation algorithms were biased towards the PPI network (Supplementary Fig. 1). Although existing drug-gene-disease node sequence generation approaches effectively produce drug and disease embeddings, the dominance of gene nodes and a small number of drug and disease nodes results in less effective representation learning.

Furthermore, current biomedKG-based drug repurposing frameworks do not utilize drug and disease similarities in a single computational framework. The concept of "guilt-by-association" (GBA)[18], where the function of a biological entity is inferred by investigating its direct neighbors, has been a cornerstone of network-based inference algorithms, including network propagation[19]. However, applying GBA to drug repurposing is not as straightforward as it is for protein function inference, where the protein and its function exist on the same layer. The reason for this is that the function of a drug is determined by its molecular-level targets, while associated diseases are based on the semantic level. To incorporate the GBA principle for drug repurposing, we suggested a "semantic multi-layer GBA" concept. The core idea of semantic multi-layer GBA is to assign the roles of drug/disease entities by simultaneously looking at their semantic neighbors, along with their topology on the biomedKG.

In an effort to overcome the hurdles of connecting drugs and diseases through a large and dense gene-gene network by employing a semantic multi-layer GBA approach, we applied teleport operation on drugs and diseases to populate paths passing drug and disease nodes. The concept of teleportation was originally introduced by the PageRank algorithm[20], which randomly teleports a walker to any node, regardless of its topology. Building upon this idea, we have extended the teleportation process to a semantic-information-guided teleportation, which associates semantically related drugs and diseases.

With this extension, we propose an algorithm that enables random walker to teleport to semantically similar drugs and diseases. Our approach is based on the premise that the semantic neighbors of drugs share biologically relevant targets with diseases, as evidenced by our preliminary network analysis results (Supplementary Fig. 2). The incorporation of a semantic neighborhood has resulted in the generation of random walk paths that provide both biological and semantic perspectives, leading to representation learning that accurately reflects the molecular and semantic contexts of entities.

Based on these ideas, we propose **DREAMwalk**, which stands for: Drug Repurposing through Exploring Associations using Multi-layer random walk. DREAMwalk incorporates semantic information guided teleportation to populate drugs and disease entities on a random walk-based path generation process. The resulting random walks are then used to create embedding vectors using a heterogeneous Skip-gram model. These embedding vectors are then used to predict drug-disease associations and repurposing probabilities using an eXtreme Gradient Boosting (XGBoost)[21] classifier. We believe the innovation of our work lies in the fact that our semantic multi-layer GBA-based technique generates a more effective embedding space that allows for mapping drugs and diseases in the same space.

Throughout this paper, we demonstrate the following contributions of DREAMwalk; (i) the expansion of a drug's role through the incorporation of additional semantic neighbor information, (ii) the generation of an embedding space that effectively captures the harmony between semantic and molecular level contexts, and (iii) the provision of interpretability of drugs' and diseases' mechanisms of action through the sampled paths. Moreover, we present case studies on breast carcinoma and Alzheimer's disease that demonstrate the potential for drug repurposing and highlight the effectiveness of the multi-layered GBA approach in uncovering novel drug-disease associations, facilitating the translation of the molecular world to actionable insights for drug repurposing.

## Results

### DREAMwalk algorithm that integrates semantic information on biomedKG for multi-layered GBA

According to the GBA principle, characteristics of a particular biological entity can be inferred by reflecting upon its neighbors. This may be the case for single-layer GBA, such as gene or protein function prediction. However, in the case of DDA, function of a drug is defined through its biological targets and MoA; yet, its indications are determined at a higher level. To infer a drug's role from both biological MoA and semantic neighbors, we introduce the concept of "semantic multi-layer GBA".

Random walk approach has been successful for shallow embedding of graphs. Random walk-based approaches first sample node sequences, and then pass them to representation learning architectures, for example, Continuous Bag-Of-Words[22] or Skip-gram[22]. Owing to its flexible and stochastic nature, the algorithm demonstrates superior performance in a number of settings[23, 24]. DREAMwalk fully utilizes this flexibility for integrating semantic level information and successfully implements the multi-layered GBA principle on biomedKG. The outline of the DREAMwalk algorithm is summarized

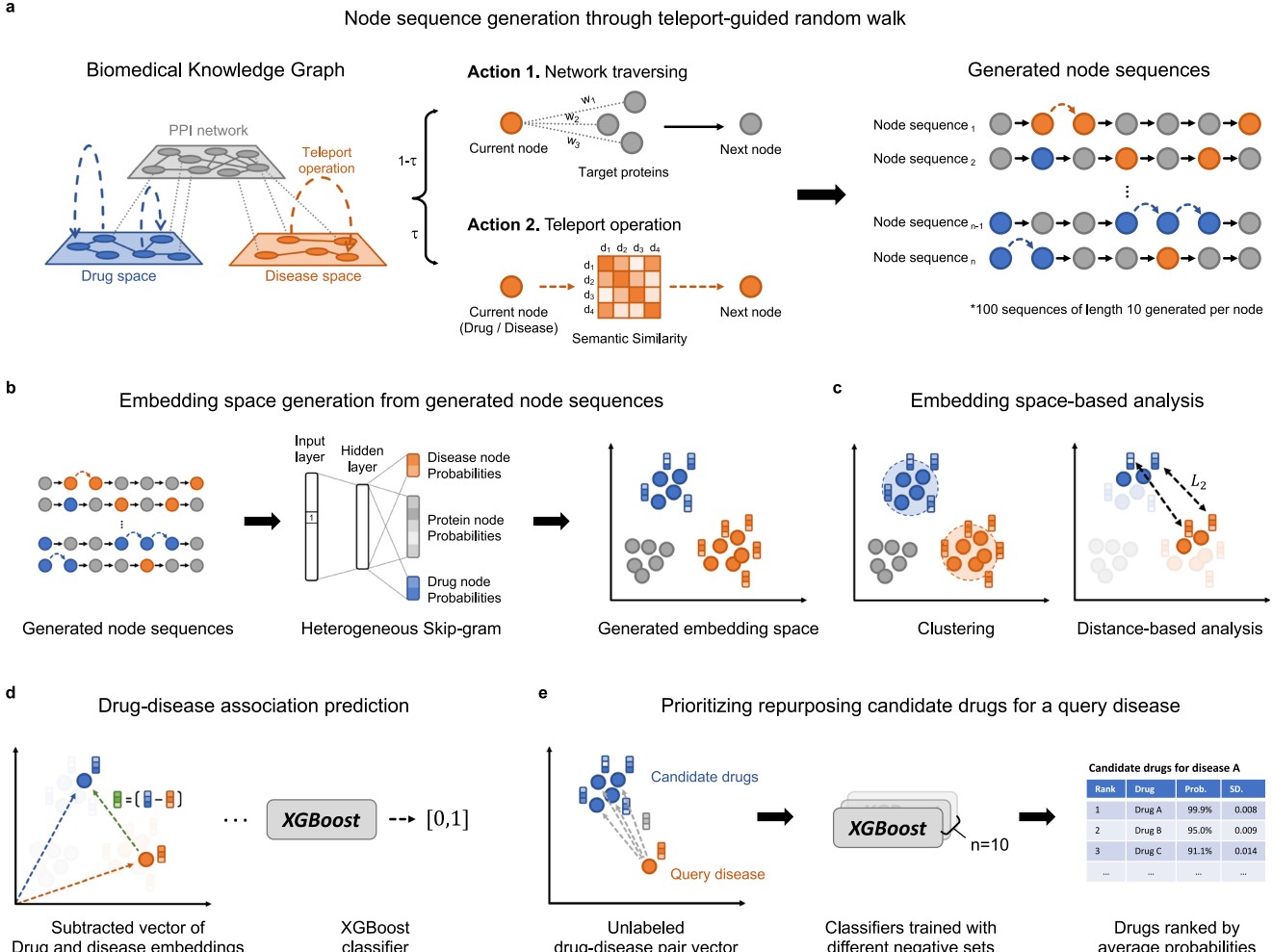

**Fig. 1 | The overview of the DREAMwalk framework. a** The node sequence generation process through teleport-guided random walk. When arriving at a drug/disease node, the random walker selects an action between network traversing and teleport operation based on the teleport factor $\tau$. **b** The embedding space generation process with heterogeneous Skip-gram model. The heterogeneous Skip-gram performs negative sampling process from the same node types. **c** The embedding vector space enables computational analysis including clustering of entities and distance-based analysis. **d** Drug-disease association prediction using XGBoost classifier with subtracted vectors of drug and disease embedding vectors as input. **e** Repurposing candidate drugs are prioritized using the trained XGBoost classifiers. Given a query disease of interest, all unlabeled drug-disease pair vectors are pass through the trained classifiers to obtain treatment probabilities. These probabilities are then averaged to yield a ranked list of candidate drugs based on their average treatment possibility.

below as illustrated in Fig. 1. Details on the technologies used in the DREAMwalk algorithm can be found in Methods Section.

1. DREAMwalk performs teleport operation while random walking by using semantic similarity as its guide (Fig. 1a). The widely used Anatomical Therapeutic Chemical (ATC) classification and medical subject headings (MeSH) describe the semantic hierarchy of drugs and diseases, respectively (Supplementary Table 1). When the random walker arrives at a drug or disease node, it selects its next action between network traversing and teleport operation. If network traversing action is selected, the random walker proceeds with network traversing procedure as it has done so far. If the selected action is teleport operation, the random walker randomly samples the next node from the similarity matrix $S_{drug}$ or $S_{disease}$, by using similarity values as sampling distribution. The probability of choosing teleport operation over network traversing is defined by the teleport factor $\tau$, which is a user-given parameter. This guided teleport operation leads the random walk sequence from the local neighborhood of the biological level network to a semantically relevant neighborhood.

2. The semantic information-integrated random walk sequences are then passed on to the heterogeneous Skip-gram model-based node representation learning. This process generates an embedding space that enables computation of the relations between entities, for example clustering or distance-based analysis (Fig. 1b,c). The results of these analyses demonstrate both semantic and biological level characteristics of the DREAMwalk's constructed embedding vector space.

3. Then, using the generated node representations, a XGBoost classifier is trained to output the drug-disease treatment probability, given the subtracted vector of drug and disease nodes (Fig. 1d). The trained XGBoost model is then utilized for drug repurposing by prioritizing highly probable treatment drug-disease relationships (Fig. 1e). Details of the model can be found in the Methods Section.

## Multi-layer GBA enables accurate prediction of drug-disease associations

Prior to drug repurposing, we first evaluated the DDA prediction performance of DREAMwalk on three biomedKGs: MSI[12], HetioNet[13], and KEGG[25]. The statistics of each network are listed in Supplementary Tables 2–4, respectively. The prediction performances were measured with prediction accuracy, area under receiver

## Drug-disease association prediction performance

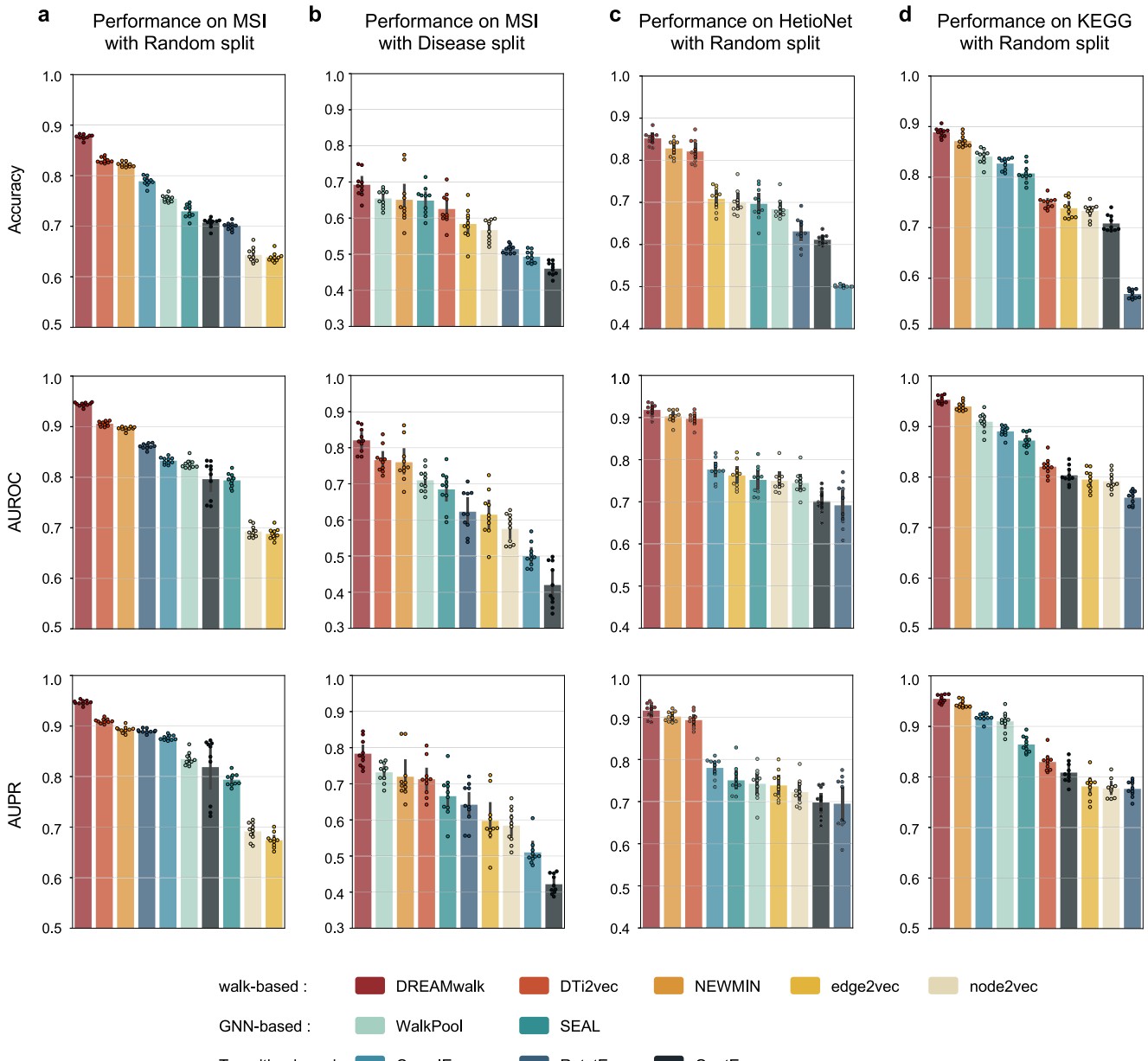

**Fig. 2 | The drug-disease association prediction performances of each model on the three biomedKGs. a** DDA prediction performance on MSI network with random split. **b** DDA prediction performance on MSI network with disease area split. **c,d** DDA prediction performance with random split on HetioNet and KEGG network, respectively. Throughout (**a**)–(**d**), The error bars denote the mean values ± 95% confidence interval, derived through $n = 10$ independent experiments. Source data are provided as a Source Data file. (AUROC Area Under the Receiver Operating Characteristics curve, AUPR Area Under the Precision-Recall curve).

operating characteristic curve (AUROC) and area under precision-recall curve (AUPR).

The selected parison models can be grouped as random walk-based model, graph neural network (GNN)-based models, and transition-based link prediction models. Random walk-based models include two similarity network-based models, NEWMIN[26] and DTi2vec[27], and non-similarity network-based models, edge2vec[28] and node2vec[29]. We also evaluated the performance of two subgraph-based GNN models, SEAL[30] and WalkPool[31], that are currently known as state-of-the-art models for link prediction tasks. Additionally, we investigated the effectiveness of transition-based models, including ComplEx[32], RotatE[33], and QuatE[34], that model relation between entities as a transition operation in the complex vector space and use score

functions to predict link probabilities. The detailed description of the methods and their model structures are in Supplementary Methods.

The results are shown in Fig. 2. In random data splitting experiments (Fig. 2a, c, d), DREAMwalk outperformed state-of-the-art link prediction models on all three biomedKGs. DREAMwalk achieved an average accuracy of 0.873, AUROC of 0.938, and AUPR of 0.939 on three biomedKG, outperforming NEWMIN, the best performing model among walk-based models with an average accuracy of 0.840, AUROC of 0.913, and AUPR of 0.913, as well as WalkPool, the best model among GNN-based models with average accuracy of 0.760, AUROC of 0.827, and AUPR of 0.829. In addition, among transition-based approaches, ComplEx achieved the best performance with average accuracy of 0.706, AUROC of 0.833, and AUPR of 0.858. Notably, the performance

comparison between proposed model and similarity-based methods, DTi2vec and NEWMIN, demonstrates that the necessity of the PPI network lies not only in biological interpretability, but also in performance improvement.

To simulate real-world drug repurposing scenarios, additional disease area-split experiment was conducted on the MSI network (Fig. 2b). The dataset was partitioned by disease categories, wherein the model must predict the treatment probabilities of drugs for unseen disease categories. This approach enables the evaluation of the model's generalizability to novel disease categories. DREAMwalk performed better over all baseline models with a margin of 6% in AUROC and 6% in AUPR, demonstrating the DREAMwalk's ability to accurately predict the potential DDA for unseen disease categories. Additionally, we conducted an experiment to compare the actual drug repurosing capabilities of the models by splitting the dataset with eight well-known repurposing cases[35] as test set. We reported the output probability of each model for the associations in Supplementary Table 7, and demonstrated that our approach achieved successful predictions for four out of eight cases at over 90% accuracy, which was the best among all tested models.

Overall, in all experiments on the three different biomedKGs MSI, HetioNet and KEGG, DREAMwalk outperformed all other comparison models in our analysis (accuracy: 0.876), ahead of state-of-the-art methods such as DTi2vec (0.840), WalkPool (0.830), and ComplEx (0.646). The integration of semantic information on biomedKG with teleport operation showed accurate and consistent prediction of DDA, along with its generalizability shown in three biomedKGs and disease split setting.

## Embedding space of DREAMwalk exhibit the harmony between biological and semantic information

We further investigated the embedding space generated by DREAMwalk to evaluate its representation of the harmonious characteristics of biological- and semantic-level contexts. Investigations with multiple perspectives were performed to evaluate the embedding space of DREAMwalk, implemented with teleport operation, by comparing it with embedding space that is constructed without teleport operation (Fig. 3a). We first observed the capability of DREAMwalk's generated space in distinguishing drug nodes from disease nodes, and also aligning drugs by their pharmacological classes (Supplementary Fig. 3). For a more detailed investigation of the generated embedding space, two case studies were performed to identify its characteristics at different levels; pharmacological and systemic pathway levels. All results reported in this section are those of the MSI network.

First, we investigated how well the embedding space represent the pharmacological level information, with hypertensive drugs as example. Hypertensive drugs from three different classes, amlodipine, labetalol and furosemide were chosen for investigation. Amlodipin is a calcium channel blocker, which lowers blood pressure through inhibiting calcium channels on the surface of vascular smooth muscle cells, leading to vasodilation[36]. Labetalol, as well as other $\beta$-blockers, treat hypertension by directly acting on the $\beta$-adrenergic receptors of the heart and reducing its stress[37]. Finally, diuretics, including furosemide, inhibit the reabsorption of ions and water in the kidney, resulting in increased diuresis and decreased blood volume[38]. As mentioned above, these three drug classes have different MoAs, hence they target different proteins, perturb different pathways and result in varying cellular events. The three drugs exhibits no interactions between their target proteins (Fig. 3b). However, they share the same disease target: hypertension, are among the first-line treatments, and are often used in combinations[39]. These characteristics of drugs with same target disease-different MoAs may be a hurdle for biomedKG-based drug repurposing.

We hypothesized that the three drugs would be located close to each other in the DREAMwalk's embedding space since their semantic roles are analogous, even though biological MoAs differ (Fig. 3c). To validate our hypothesis, normalized Euclidean distances between the three drugs were measured on both DREAMwalk-generated space (with teleport) and space generated without teleport. Each space was generated for ten times with different random seeds because the random walk algorithm is stochastic. As shown in Fig. 3e, the measured distances of three pairs display significant reduction with the integration of semantic-level information. Since the three drugs with different MoAs but same disease targets are located closer in the multi-layer embedding space, the GBA principle can be applied in a more reasonable way for drug repurposing.

The next example demonstrates the systemic pathway level information implied in the embedding space. Enalapril and valsartan are drugs that target proteins in the same hormone system, known as renin-angiotensin-aldosterone system (Fig. 3d). Enalapril is an angiotensin-converting enzyme inhibitor, and valsartan is an angiotensin receptor blocker. Since the MoAs of the two drugs exist in the same system, they both treat hypertension, and are clinically recommended to be used separately as their combination is associated with adverse effects[40, 41]. However, in the biomedKG, they do not share targets. In addition, the Jaccard similarity of the PPI neighbor set of ACE and AGTR1 was 0.018 (Fig. 3c), implying a notable biological distance between the two drugs.

The measured normalized Euclidean distance of the two drugs significantly decreased with the integration of semantic-level information (Fig. 3f). Their cellular pathways are well implied in the embedding space constructed without semantic teleport since the biomedKGs already contain pathway or molecular function entities. However, biological system level pathway interactions, for example, hormone systems, do not appear to be sufficiently contained in biomedKGs. Hence, our case study demonstrates the practicality of the multi-layer GBA approach in narrowing down this gap between molecular and systemic levels.

For a quantitative demonstration of the decrease of Euclidean distance of drugs sharing same indications, we measured all-pairwise distance between the treatments for four disease with the most number of treatments; rheumatoid arthritis (92 drugs), asthma (88 drugs), hypertensive disease (82 drugs), and allergic rhinitis (74 drugs). The all-pairwise distance distribution of the four disease-curing drugs also demonstrates the significant decrease with semantic information-guided teleport. Overall, the two examples suggest that embedding space of DREAMwalk, generated with semantic information as guide, reflects entities' relations at different levels, thereby locating therapeutically associated entities closer.

## semantic information-guided teleport is essential for performance improvement in DREAMwalk

DREAMwalk's multi-layer GBA strategy, which integrates semantic information into biomedKGs, is implemented through a teleport-guided random walk algorithm. To demonstrate the efficiency and significance of semantic information-guided teleport operation in accurate DDA prediction, we conducted several ablation studies. Integrating semantic information, such as semantic hierarchies of drugs (ATC classification) and diseases (MeSH term, Disease Ontology, ICD-11) is a key principle of the DREAMwalk framework. To compare the model performances with equal amounts of information for baseline models, we integrated semantic hierarchies as nodes on the biomedKGs. Figure 4a illustrates the network learning with hierarchy entity nodes attached, in comparison with teleport operation.

First, we compared the DDA prediction performances of three models: model without teleport, model with hierarchy as nodes, and model with semantic information-guided teleport on the MSI network. The experimental results show that the integration of hierarchical information as nodes indeed produced a more accurate prediction (Fig. 4b). Notably, the increase in AUPR and AUROC was significantly

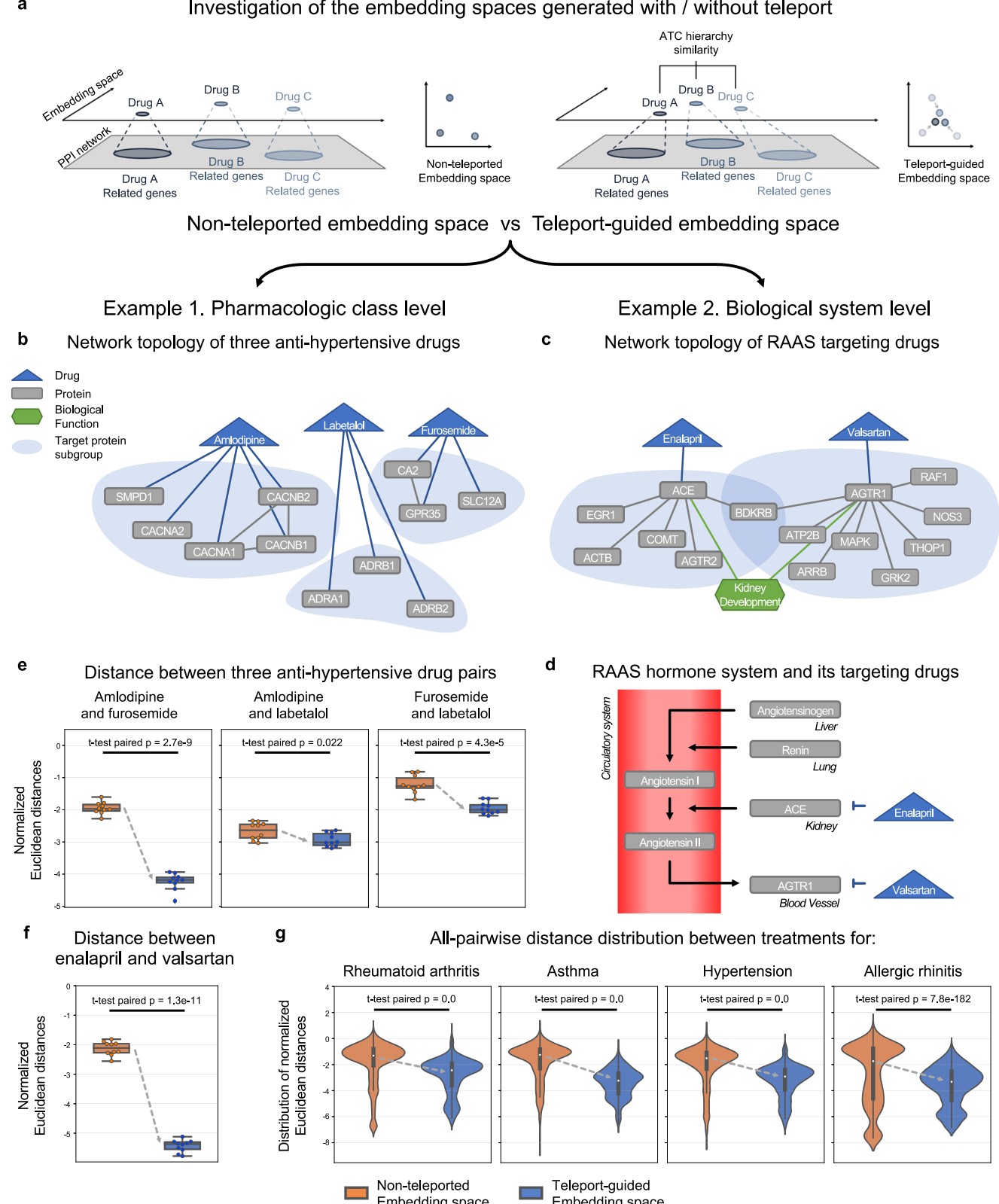

**Fig. 3 | The embedding space of DREAMwalk reflects the pharmacological and biological system-level characteristics of drugs. a, b** Network topology of the three hypertensive drug classes on without-teleport embedding space (left) and DREAMwalk embedding space (right). **c, d** Network of RAAS and its two targeting drugs. **e** The normalized euclidean distance between the hypertensive drug pairs on DREAMwalk embedding space (blue) and without-teleport embedding space (orange). **f** The normalized euclidean distance between the RAAS targeting drugs. On boxplots of (**e**) and (**f**), the center line represents the median, while the upper and lower box limits represent the quartiles. The whiskers indicate 1.5 times the interquartile range. All data have been derived through $n = 10$ independent experiments. **g** The all-pairwise normalized euclidean distance distribution for all drug treatments for rheumatoid arthritis, asthma, hypertension and allergic rhinitis. On the violin plot, the white dot represents the median, while the thick bar represents the interquartile range and the thin line indicates 1.5 times the interquartile range. Source data are provided as a Source Data file. (RAAS Renin-Angiotensin-Aldosterone System, t-test paired two-sided paired t-test).

higher with semantic information-guided teleport (AUROC: 0.944, AUPR: 0.947) compared with using the semantic information alone as network components (AUROC: 0.922, AUPR: 0.929).

There are advantages of integrating semantic-level information through teleport operation over learning them as network components. Along with computational efficiency due to the smaller node and edge counts (Supplementary Table 1), the signal-to-noise (S/N) ratio may be controlled by applying a cut-off value to the similarity matrix. Introducing the whole hierarchy as a network component possesses the limitations of an uncontrollable S/N ratio. Applying a cut-off or threshold value for similarity matrix construction can effectively reduce the number of edges (Fig. 4c). This approach not only narrows the search space but also eliminates noise resulting from irrelevant neighbors. The performances of the DREAMwalk model based on different similarity cut-off values are shown in Fig. 4d. The performance increased as cut-off increased until 0.4, implying a decrease in S/N ratio as dissimilar entities are excluded from teleportable neighbors. Notably, we also observed varying levels of performance improvements and decreases among the baseline models following the addition of hierarchy nodes (Supplementary Fig. 4). These findings suggest that the semantic hierarchy information may act as a guide if appropriately utilized; otherwise, it may introduce noise into the network, leading to performance degradation.

In addition, teleport factor $\tau$ can be modified for balancing representation learning between biological and semantic levels and controlling S/N ratio (Fig. 4e). The capability of DREAMwalk algorithm in setting the cut-off point to the optimal value and maintaining the S/N ratio makes it more powerful tool in accurately predicting DDAs.

Another study was performed to investigate the contribution of semantic information towards performance enhancements. As previously mentioned, the biomedKGs are heavily biased to the PPI network, since the number of nodes and their degree are much higher than the other components (Supplementary Fig. 1). We hypothesized that the teleport operation's nature of populating drug- and disease-passing paths may have contributed the most to the improved performance by debiasing the biomedKG learning process from PPI. To determine whether the use of semantic information contributed to the increase in performance, an additional experiment is conducted by performing teleport operation randomly. When random teleport-guided random walker selects its action as teleport, it selects the next node from randomly generated transition weights instead of semantic similarity matrix as transition weights. The random generation of the transition matrix was performed for 10 times, and the resulting performances compared with models without teleport and with semantic information-guided teleport are provided in Fig. 4b.

The results show the performance of random teleport model (AUROC: 0.885, AUPR: 0.893) significantly lower than that of the without-teleport model (AUROC: 0.905, AUPR: 0.912), let alone the semantic information-guided teleport model (AUROC: 0.944, AUPR: 0.947). This leads to the conclusion that a semantically relevant guide is necessary for teleport operation to exert its potentials, and using both adequately results in synergistic improvement in drug-disease association prediction.

### DREAMwalk's semantic information-guided path enables interpretation of drug/disease mechanisms

Use of biomedKGs for learning and predicting drug-disease associations offers interpretability, compared to alternative black-box learning methods. The node sequences generated by DREAMwalk can be analyzed to identify neighboring genes for a given entity. In this section, we demonstrate an approach for inferring the MoA of a biomedical entity based on neighboring genes. For the functional analysis of neighboring genes, we first defined a 'window neighbors' of an entity as the set of genes within a window of given size $l$ in the generated node sequences (Fig. 5a).

Teleport-guided random walk of DREAMwalk is expected to not only explore the local neighborhood of the PPI network but also broaden the search range to semantically relevant regions. We first observed that teleport introduces more diversity to window neighbors compared with non-teleported paths (Supplementary Fig. 6).

Gene set enrichment analysis was further performed to demonstrate the biological interpretability of teleport-guided neighborhoods in explaining drug and disease mechanisms compared to non-teleported neighbors. Case studies were investigated with drug "gabapentin" and disease "Parkinson's disease (PD)". Window neighbors were selected from the window of size $l = 2$. The enrichment analyses were performed using Enrichr[42]. Gabapentin is a relatively novel drug used in the treatment of epilepsy. The effects of gabapentin on brain neurotransmitters, including gamma-aminobutyric acid (GABA), have yet to be elucidated. Studies have reported that gabapentin significantly increases GABA levels in the brain[43, 44]. Interestingly, even though gabapentin alters and structurally mimics GABA, the drug does not seem to directly affect GABA-specific enzymes or receptors[45]. Drug-target databases reflect that gabapentin does not directly bind to GABA receptors[46, 47]. GO Molecular Function (MF)-enrichment was performed to examine the window neighboring genes of gabapentin on both teleport-guided and non-teleported paths. The resulting top 20 MFs based on adjusted p-values are shown in Fig. 5b. As mentioned, gabapentin does not directly target GABA receptors, so GABA-related proteins are not located close to gabapentin in the biomedKG. Because the non-teleport neighbor set is generated based on local neighbors of gabapentin, GABA-related MFs are not in the high ranks of its enrichment results. In contrast, teleport-guided neighbor set captures GABA-related MFs. Enriched MFs that appeared only in the top 20 MFs of the teleport neighbors included GABA receptor activity (adj. $P = 4.30E-19$), GABA-gated chloride ion channel activity (adj. $P = 1.92E-17$), and GABA-A receptor activity (adj. $P = 1.39E-16$). Gabapentin's GABAergic activities, although not contained in drug-target interactions, are well captured through the semantic information-integrated embedding space of DREAMwalk.

Parkinson's disease (PD), one of the most common neurodegenerative diseases in the elderly, mainly occurs due to depletion of neurotransmitter dopamine. KEGG enrichment was performed using the window neighbors of PD (Fig. 5c). Among the top 20 enriched KEGG pathways in both neighbor sets, the pathways that appeared only on DREAMwalk were Apoptosis pathway (adj. $P = 2.36E-19$), Fluid shear stress and atherosclerosis pathway (adj. $P = 2.18E-18$) and Focal adhesion (adj. $P = 4.10E-18$). Literature validation confirmed that these pathways are closely related to PD. Apoptosis is regarded as one of the main mechanisms of neuronal death in PD[48]. Although the specific processes of PD are not completely understood, it has been observed that these convergent mechanisms result in neuronal death through apoptosis[49, 50]. Fluid shear stress and atherosclerosis pathway, along with lipid and atherosclerosis pathway (adj. $P = 2.27E-21$) represent the association between PD and atherosclerosis. Several studies have supported the association between atherosclerosis and PD, as well as other neurodegenerative disorders[51, 52]. A large-scale Atherosclerosis Risk in Communities study[53]-based analysis reported that decreased heart rate variability, a well-known cause of fluid shear stress and atherosclerosis[54], was associated with an increased risk of PD[55]. Finally, Focal adhesion pathway is known to be associated with PD because adhesion plays a role in neuroprotection[56] and the structure and function of the synapses[57]. A genome-wide association studies (GWAS) and gene expression-based integrative studies have also reported Focal adhesion as a consensus disease pathway in PD[58].

The path-based interpretation of gabapentin and PD show the semantic information integrated neighborhoods' potential ability to explain biological mechanisms of drugs and diseases, which are difficult to identify solely via molecular-level neighborhoods.

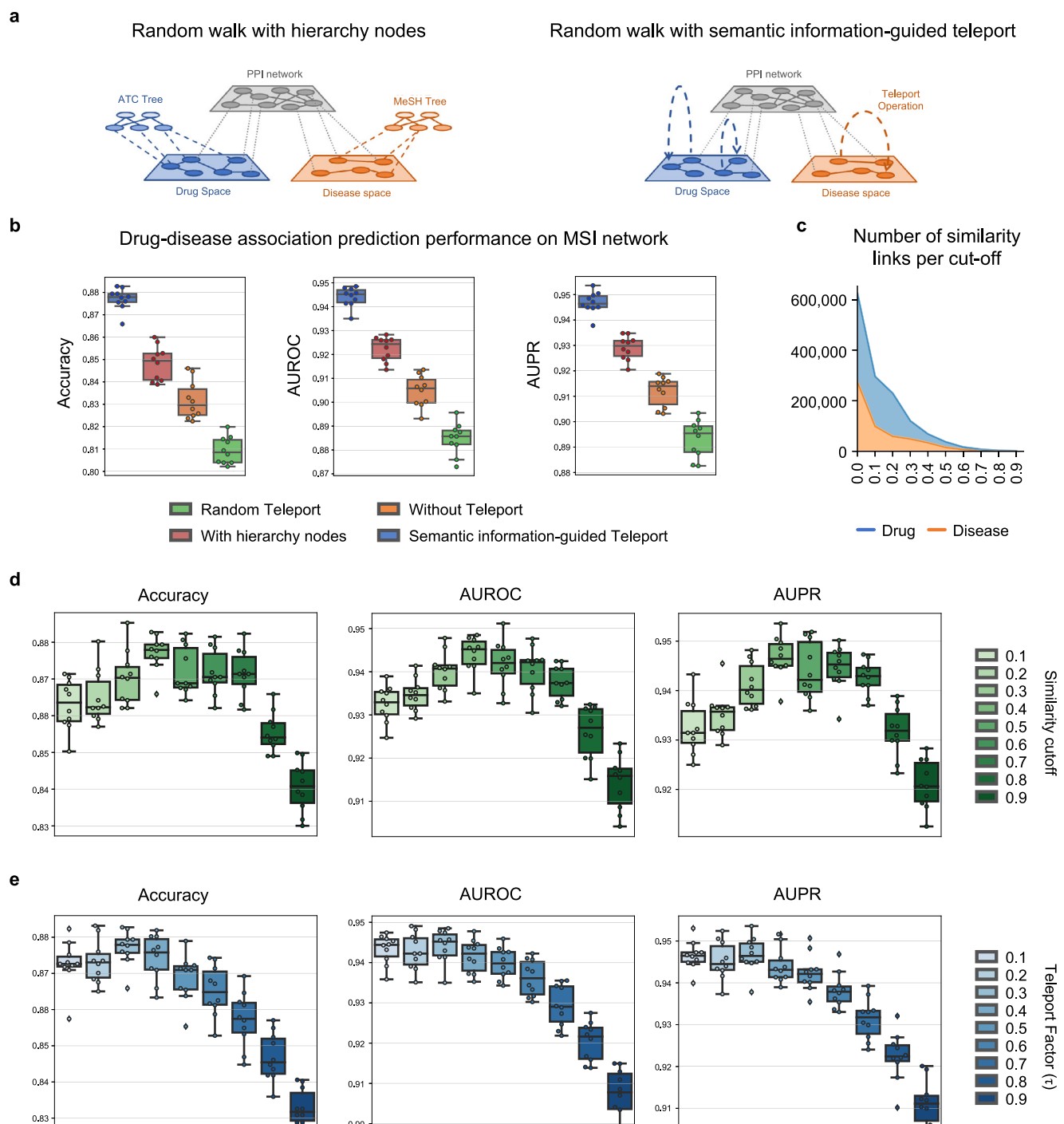

**Fig. 4 | Ablation study results of DREAMwalk's teleport operation. a** Concept illustration of Random walk with hierarchy nodes (left) and semantic information-guided Teleport (right). **b** Drug-disease association (DDA) prediction performances of models random teleport (green), without teleport (orange), with hierarchy nodes (red) and semantic information-guided teleport (blue) on MSI network. **c** Stacked area plot of number of similarities of drug (blue) and disease (orange) per cut-off. **d** DDA prediction performances following the change in similarity cut-off.

Teleport factor was fixed at 0.3. **e** DDA prediction performances following the change in teleport factor $\tau$. Similarity cut-off was fixed at 0.4. On box plots of (**b, d, e**), the center line represents the median, while the upper and lower box limits represent the quartiles. The whiskers indicate 1.5 times the interquartile range. All data have been derived through $n = 10$ independent experiments. Source data are provided as a Source Data file. (AUROC Area Under the Receiver Operating Characteristics curve, AUPR Area Under the Precision-Recall curve).

## DREAMwalk suggests potential repurposable drugs for Alzheimers' disease and breast cancer

As our goal is to suggest the repurposing use of existing drugs, repurposing candidate drugs were selected for breast carcinoma and Alzheimer's disease (AD) on the MSI network. For each disease, drug-disease association probabilities for all unlabeled drugs were calculated ten times using DREAMwalk trained with different negative sets. After calculating the average probabilities, top 10 high-probable drugs were selected as candidates for drug repurposing. The average probabilities and their standard deviations (SD) for each drug are listed in Table 1, along with their original indications and supporting literature for repurposing evidence.

**a**

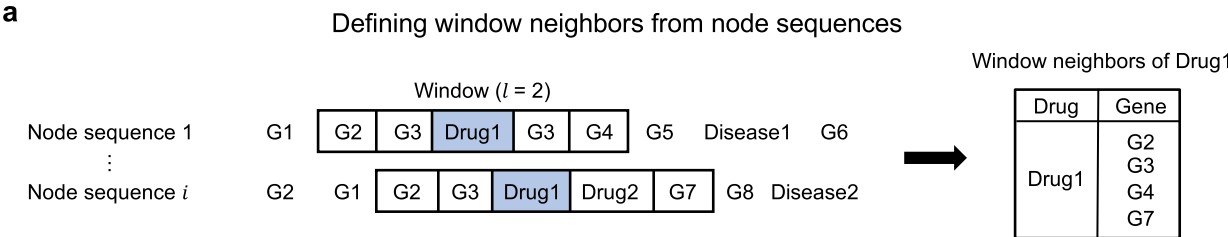

**b**

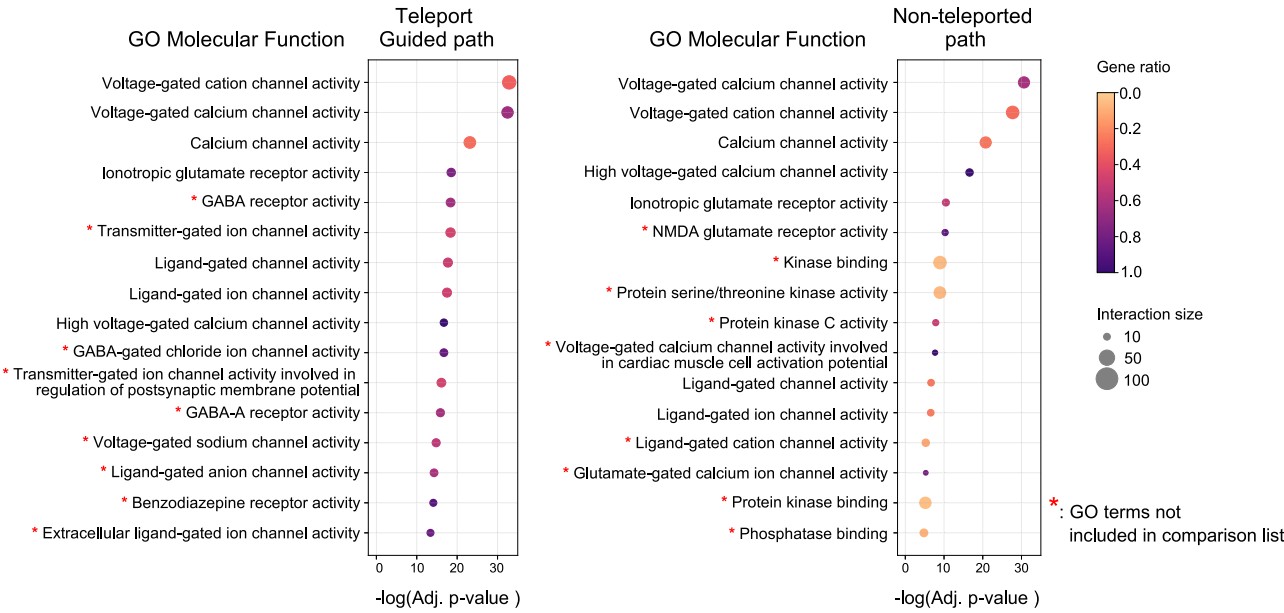

**c**

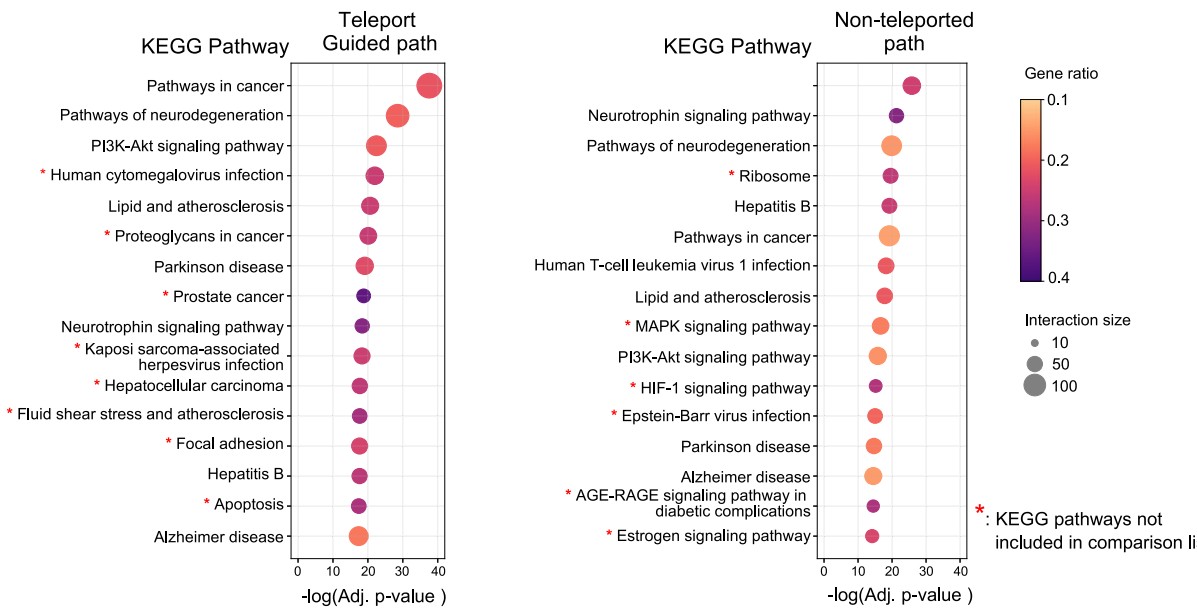

**Fig. 5 | The window neighbor gene set analysis results. a** Selection of Drug1's window neighbor genes from node sequences using window of length 2. **b** GO ontology enrichment results of window neighbors of drug "gabapentin". **c** KEGG enrichment results of window neighbors of disease "Parkinson's disease". Source data are provided as a Source Data file. Fisher's Exact test and Benjamini-Hochberg method have been applied for calculating the adjusted p-values. (Adj.: Adjusted).

**Table 1 | Drug repurposing candidates of DREAMwalk for breast carcinoma and Alzheimer's disease**

| Rank | Drug | Original Indication | Avg. prob. | SD | Evidences |
|---|---|---|---|---|---|
| **Breast Carcinoma** | | | | | |
| 1 | Irinotecan | Colorectal cancer, SCLC, NSCLC | 0.995 | 0.0033 | 59, 81, 82 |
| 2 | Etoposide | Germ cell tumors, Kaposi sarcoma, SCLC | 0.993 | 0.0066 | 59, 60 |
| 3 | Dactinomycin | Wilm's tumor, Rhabdomyosarcoma, Neuroblastoma | 0.992 | 0.0128 | 83 |
| 4 | Teniposide | ALL, Small Cell Carcinoma, Lymphoid Leukemia | 0.991 | 0.0109 | 61, 62 |
| 5 | Vinblastine | Hodgkin disease, Lymphoma, NHL | 0.989 | 0.0125 | 59, 60 |
| 6 | Mitoxantrone | AML, Multiple Sclerosis, Lymphoma, Sarcoma | 0.989 | 0.0128 | 60 |
| 7 | Interferon alfa-2b | Melanoma, Brain Neoplasms, Hepatitis C | 0.988 | 0.0095 | - |
| 8 | Cortisone-acetate | SLE, CTCL, IBD, Autoimmune Diseases | 0.987 | 0.0218 | - |
| 9 | Vindesine | CML, Melanoma, ALL, Hodgkin's lymphoma | 0.987 | 0.0100 | 63, 64 |
| 10 | Hydroxyurea | CML, cancer of head and neck, sickle cell anemia | 0.986 | 0.0128 | 84–87 |
| **Alzheimer's disease** | | | | | |
| 1 | Levetiracetam | Simple partial seizures, Epilepsy | 0.990 | 0.0137 | 67, 68 |
| 2 | Clomipramine | OCD, Chronic pain, Narcolepsy | 0.983 | 0.0161 | 88, 89 |
| 3 | Duloxetine | Major depressive disorder, Peripheral neuropathy | 0.977 | 0.0236 | 65 |
| 4 | Fluoxetine | Major depressive disorder, OCD, Bipolar disorder | 0.976 | 0.0315 | 65, 90–93 |
| 5 | Maprotiline | Depressive disorder, Duodenal Ulcer | 0.974 | 0.0253 | - |
| 6 | Armodafinil | Narcolepsy, Obstructive sleep apnea | 0.974 | 0.0278 | 94 |
| 7 | Sertraline | Depressive disorder, OCD, Panic disorders | 0.973 | 0.0351 | 65, 95–97 |
| 8 | Lisdexamfetamine | ADHD | 0.971 | 0.0305 | 98 |
| 9 | Atomoxetine | ADHD | 0.968 | 0.0509 | 66, 99 |
| 10 | Dextroamphetamine | ADHD, narcolepsy | 0.967 | 0.0392 | 100 |

*Avg. prob.* average probability; *SD* standard deviation; *ADHD* attention-deficit/hyperactivity disorder, *ALL* acute lymphocytic leukemia; *AML* acute myelocytic leukemia; *CML* chronic myeloid leukemia; *CTCL* cutaneous T-cell lymphoma; *NHL* non-Hodgkin lymphoma; *NSCLC* non-small cell lung cancer; *OCD* obsessive compulsive disorder; *SCLC* small cell lung cancer; *SLE* systemic lupus erythematosus.
The drug-disease association probabilities were measured ten times, and the average and the SD of the predicted probabilities are provided in the table.

The top ten repurposing candidates for breast carcinoma included various chemotherapeutic agents, some of which are frequently administered off-label for treating metastatic breast cancers and other metastatic cancers in clinical settings[59, 60]. Additionally, the list comprised drugs that were more widely employed for treating breast cancer during the 1980s, but have since seen reduced usage[61–64], setting them absent from the current treatment list for breast cancer in the BiomedKG. A notable observation was the inclusion of Interferon alfa-2b and Cortisone-acetate in this list, suggesting that the model recognizes the relationship between breast cancer treatment and an immunological approach (in the case of Interferon alfa-2b) or the ability of glucocorticoids to mitigate chemotherapy agents' side effects (in the case of Cortisone-acetate).

AD is one of the most common causes of dementia, a neurodegenerative disorder of the cerebral cortex and limbic system that results in mild cognitive decline and memory loss. Among the ten high-probable repurposing candidates, there were four anti-depressive drugs and five ADHD/narcolepsy treatments, supported by multiple clinical and molecular evidences. The relationship between depression and AD progression[65], as well as the link between cognitive-enhancing drugs and AD[66], has been highlighted by previous research. An interesting case was the top-ranked drug levetiracetam, an anti-epileptic drug, which has been shown to improve spatial memory in a recent randomized clinical trial[67]. Although it lacks a shared target with AD or any other known AD treatments, our model successfully identified the potential of levetiracetam in improving the symptoms of AD patients with the epileptic variant.

Furthermore, the top ten lists of baseline models for both breast carcinoma and AD are provided in Supplementary Table 5. Comparing these lists highlights DREAMwalk's capability to identify a related yet diverse range of repurposable drugs when compared to the baseline models. Also, the bottom-ten list of DREAMwalk listed drugs from therapeutically unrelated areas for breast carcinoma and AD (Supplementary Table 6).

Additionally, for clinical interpretation of prediction results, we examined the models' performance in predicting DDAs for AD repurposing candidates in phase 3 clinical trials as of 2021[68] (Table 2). Our study focused on eight drugs: Brexpiprazole, Caffeine, Escitalopram, Guanfacine, Hydralazine, Metformin, and Omega-3-carboxylic acids, which exists in the MSI network. For each model, we predicted the probability for all unlabeled drugs for ten times and ranked them based on their average probabilities. Our model predicted the DDAs with the highest median probability and rank compared to the seven baseline models. Specifically, two drugs (Caffeine and Escitalopram) had a predicted probability of over 0.8, indicating a strong likelihood of a drug-disease link, which no other models were able to output but once for DTi2vec. Also, DREAMwalk had the highest predicted probability among all models for three drugs (Brexpiprazole, Caffeine, and Guanfacine).

In summary, literature-based evaluation from in vitro experiments to clinical case reports and off-label uses demonstrated the potential repurposability of our candidate drugs for breast carcinoma and AD. Especially, AD drug repurposing candidates in phase 3 clinical trial showed the highest median probability and rank compared to all baseline models. Overall, the presented results and case studies demonstrate the usefulness of DREAMwalk in deriving new hypotheses for DDAs that would facilitate experimental and clinical validation and ultimately provide novel treatment strategies for treatment-poor diseases.

## Discussion

The DREAMwalk framework implements a semantic multi-layer GBA for accurate DDA prediction and drug repurposing by introducing the semantic neighbors of drug and disease entities. By integrating semantic information-guided teleport technique to the random walk algorithm, our representation learning process incorporates both molecular- and semantic-level information and generates a harmonized embedding space of drugs and diseases. The high DDA

**Table 2 | Predicted probabilities and rankings of repurposing candidate drugs in phase 3 clinical trial for Alzheimer's disease in 2021**

| Models | Brexpiprazole | Caffeine | Escitalopram | Guanfacine | Hydralazine | Metformin | Omega-3-carboxylic acids | Median |
|---|---|---|---|---|---|---|---|---|
| DREAMwalk | 0.736 (210) | **0.910** (59) | **0.875 (82)** | **0.602 (326)** | 0.481 (412) | 0.03 (1440) | 0.342 (564) | **0.602 (326)** |
| edge2vec | 0.499 (1496) | 0.582 (975) | 0.616 (738) | 0.584 (962) | 0.626 (665) | 0.531 (1362) | **0.592** (912) | 0.584 (962) |
| ComplEx | 0.466 (886) | 0.680 (**49**) | 0.412 (1180) | 0.503 (698) | **0.678 (51)** | **0.531 (549)** | 0.409 (1199) | 0.503 (698) |
| DTi2vec | **0.769** (211) | 0.706 (238) | 0.870 (104) | 0.11 (970) | 0.11 (968) | 0.047 (1346) | 0.443 (456) | 0.443 (456) |
| node2vec | 0.338 (1631) | 0.556 (204) | 0.439 (982) | 0.477 (606) | 0.508 (366) | 0.350 (1618) | 0.404 (1327) | 0.439 (982) |
| NEWMIN | 0.692 (**120**) | 0.576 (273) | 0.596 (248) | 0.431 (617) | 0.391 (925) | 0.380 (1037) | 0.436 (589) | 0.436 (589) |
| WalkPool | 0.237 (1471) | 0.62 (179) | 0.227 (1538) | 0.239 (1458) | 0.436 (514) | 0.386 (747) | 0.509 (**292**) | 0.386 (747) |
| SEAL | 0.338 (1085) | 0.518 (247) | 0.426 (699) | 0.281 (1396) | 0.337 (1086) | 0.341 (1071) | 0.363 (980) | 0.341 (1071) |

For each model, the drug-disease association probabilities were measured for ten times, then ranked by their average probabilities. The ranks of the drugs are shown in parentheses, and the highest probability and rank for each drug are highlighted in bold.

prediction performance on the three biomedKGs of MSI, HetioNet and KEGG demonstrates the generalizability of teleport-mediated integration of semantic and biological information. Ablation studies support this concept by demonstrating that semantic information-guided teleport is essential for prediction performance enhancement. semantic information injected semantic similarity measure provides the largest performance enhancement, whereas randomly performed teleport results in poor performance.

The high performance of DREAMwalk on DDA prediction is based on the effective embedding space of drugs and diseases that are constructed from semantic information-guided random walks. The examples on hypertensive drugs demonstrated the pharmacological- and biological system-level information reflected in the embedding space. In addition, the generated paths offer interpretability of drug and disease associations. The gene set enrichment analysis on gabapentin and PD with the sampled paths displayed the interpretability of DREAMwalk in determining the mode of actions of drug and disease. Finally, DREAMwalk's predicted repurposing candidate drugs for breast carcinoma and AD are well supported in literature.

There are however some potential limitations to DREAMwalk's current DDA prediction and drug repurposing framework. Although teleport operation offers efficient integration of semantic information into the biomedKG representation learning process, the teleport probability $\tau$ is a user-specified hyperparameter and $\tau$ is fixed to a preset value throughout the entire network. Adaptation of teleport probability based on local network topology may offer more flexible integration of semantic data. In addition, downstream DDA prediction task of DREAMwalk is trained based on randomly sampled negative drug-disease pair, owing to the lack of public data on negative drug-disease pairs, we plan to develop an adequate positive-unlabeled learning framework for more accurate DDA prediction with more reliable decision boundaries in future studies.

In summary, our results indicate that the multi-layer GBA principle can be used for more accurate computational drug repurposing, inferring from semantic neighbors via random walk with semantic information-guided teleport. We believe that our work is a demonstration of how to efficiently leverage semantic information in machine learning frameworks on biological domain, as adequate integration of different levels of information is key to translating molecular information to the clinical world. Also, our work may provide clues for pharmaceutical scientists to discover effective treatments for diseases that are currently without treatment options.

## Methods
### Biomedical knowledge graphs
Three biomedKGs, MSI[12], HetioNet[13], and KEGG[25], were used for evaluating the drug-disease link prediction performances of the proposed and baseline models. Each biomedKG consists of varying types of nodes and edges. Node types other than drug, disease, gene and

pathway (or biological function) were eliminated to construct a molecular-level biomedKG. Associations of higher level, for instance adverse effect or anatomy, were excluded during this process. Also, during node embedding generation step, all the drug-disease treatment edges were removed from the biomedKG. This allows the node representation learning process to fully incorporate the biological and semantic contexts of entities, without treatment association information. The drug-disease association pairs were later used for downstream tasks of XGBoost classifier-based DDA prediction.

**Multi-scale interactome (MSI) network**. Multi-scale interactome (MSI) network[12] is a multi-scale heterogeneous biomedKG, including not only molecular-scale interaction but also their functional annotations. After constructing a multi-scaled biomedKG of drug, disease, protein and Gene Ontology[69] Biological Function nodes, the authors generated diffusion profiles for each node through weighted network propagation and performed downstream analyses, e.g., drug mechanism analysis and drug-disease association prediction. The original MSI network consists of four node types and four edge-types. Since the network consists of only drug, disease, protein and GO biological function, the whole network is utilized for experiments and drug repurposing procedure of this work. The statistics and the data source of node and edge information are provided in Supplementary Table 2.

**HetioNet**. HetioNet[13] is a biomedKG of 11 types of nodes and 24 edge types, from 29 publicly available data sources. HetioNet is designed to integrate every available resource into a single interconnected data structure to assess the systematic mechanisms of drug efficacy. To use the original network to our experimental settings, node types other than drug, disease, protein (gene) and pathway were removed. The original and the processed network statistics are provided in Supplementary Table 3. It is worth mentioning the HetioNet contains a smaller number of disease nodes compared to MSI and KEGG because the disease nodes defined are at a higher or broader level. For instance, all hypertensive disorders and its relationships are summarized into single "hypertension" node in HetioNet, while the MSI network contains not only "Hypertensive disease" but also variations of the disorders, e.g., "Intracranial Hypertension", "pulmonary arterial hypertension (PAH)", "ocular hypertension", and more.

**KEGG**. KEGG[25] is one of the most widely-used database of expert-curated molecular- and pathway-level interaction annotations. The whole KEGG database contains 15 sub-databases of different types of entities. The systems information is contained in PATHWAY, BRITE and MODULE sub-databases, and the genomic information is contained in a latter-developed KEGG Orthology (KO) database. KEGG DISEASE database contains information of disease entities and their relationship to disease genes, carcinogens, pathogens and other

environmental factors. KEGG DRUG database of approved drugs lists information of drug target information along with drug metabolism information. Of all the relationships contained in 15 sub-databases, only gene-pathway, drug-gene and disease-gene relationships were utilized. The statistics of the utilized KEGG network is provided in Supplementary Table 4.

## Drug and disease semantic similarities
Integrating semantic level information to biomedKG of drug-gene-disease enables drug-disease association prediction through multi-layer GBA perspective. To leverage the tree-structured hierarchical annotation of drug and disease nodes, DREAMwalk utilizes semantic similarity measure as teleport probability between drug-drug or disease-disease nodes. Based on public drug and disease ontologies, an information content-based semantic similarity measure was adopted for calculating the semantic similarities of drug-drug and disease-disease pairs. The detailed process for calculating the similarities are described below.

**Drug and disease ontologies.** The utilized drug hierarchy is ATC classification hierarchy for all three biomedKGs. The ATC codes were assigned to drugs using the information provided by Drugbank[46]. Since the diseases IDs were mapped to different hierarchies in each biomedKG, two different disease hierarchies were used; Hetio's disease entities were mapped to Disease Ontology[70] hierarchy, MSI diseases were mapped to Medical Subject Heading (MeSH)[71] term hierarchy, and KEGG diseases were mapped to ICD-11[72]. All the hierarchies can be regarded as a directed acyclic graph of terms.

**Information content.** A number of measures have been proposed for calculating the similarity of entities in biomedical ontologies since the 1990s[73–75]. Some measures compare the entities' information content (IC) when measuring their similarity. IC gives a measure of how informative an entity $c$ is, based on the occurrence frequency of an entity in a given biomedical corpus, e.g., Uniprot Knowledge base[76]. More frequent an entity appears, less informative it is, so smaller IC is assigned to the entity. Calculating the IC value of an entity directly from a tree-structured hierarchy instead of a given corpus can be performed through counting $N_{child}$, which is the number of children a term has in the hierarchy structure, as proposed by ref. [77]. The IC value of a term in a hierarchy structure can be defined as following:

$$IC(c) = 1 - \frac{\log(N_{child}(c) + 1)}{\log(N_{child}(root))} \qquad (1)$$

The denominator of the equation (1) assures the IC values are in [0,1], and the information content of the top entity is equal to 0.

**Semantic similarity.** Among the most commonly used semantic similarity measures[73–75], DREAMwalk adopted the semantic similarity measure proposed by ref. [73]. According to the authors, given the IC value of two entities $c_1, c_2$ and their Most Informative Common Ancestor, the distance between the two entities can be defined as following:

$$dist(c_1, c_2) = IC(c_1) + IC(c_2) - 2 \times IC(MICA(c_1, c_2))$$

Since $max(IC) = 1$, the maximum value of the semantic distance between two entities is 2. In order to transform the distance measure into similarity value in range of [0,1), the similarity measure can be defined as below:

$$sim(c_1, c_2) = 1 - \left(\frac{dist(c_1, c_2)}{2}\right) \qquad (2)$$

Using Eq. (2), similarity measure is calculated all-pairwise for drugs and disease of the three biomedKGs, according to their drug/disease hierarchies. This procedure returns a similarity matrix $S \in \mathbb{R}^{n \times n}$ where $n$ is the number of drug or disease contained in a biomedKG. A user-defined cutoff may be introduced for eliminating the pair information with similarity below the given cutoff value for the reduction of noise and improvement of computational efficiency. For our study, the cutoff is empirically set to 0.4 for all biomedKGs and all similarities below are masked.

## Multi-layered GBA through teleport-guided random walk
Implementing the multi-layer GBA concept requires the introduction of semantic neighbors on network feature learning frameworks. To introduce the semantic-level information on molecular-level biomedKGs, we augmented a random walk algorithm with semantic information-guided teleport operation, which is inspired from the PageRank algorithm[20].

The teleport-guided random walker generally traverses the biomedKG by following its edges; however, when it arrives at drug/disease nodes, it randomly selects an action between teleport operation and network traversing based on the user-given teleport factor. If the selected action is the teleport operation, the random walker teleports to a randomly sampled node based on the similarity matrix $S$. Otherwise, if the action is network traversing, the random walker resumes the traversing process. For all nodes in each network, 100 walks of length four were sampled. The detailed algorithm is provided below.

**Random walk.** The random walk algorithm traverses nodes in the network, generating a node sequence $p = n_1, n_2, \ldots, n_l$ that can be used in the heterogeneous Skip-gram based graph learning framework. A node sequence of length $l$ from a network $G = (V, E)$ of node set $V$ and edge set $E$ can be generated by following distribution:

$$P(n_i = x \mid n_{i-1} = v) = \begin{cases} \pi_{vx}/Z & \text{if } (v,x) \in E \\ 0 & \text{otherwise} \end{cases}$$

where $\pi_{vx}$ is the unnormalized transition probability between nodes $v$ and $x$, with $Z$ as a normalizing constant. The transition probability of the unbiased random walk introduced in word2vec[22] is equal to the edge weights $w_{vx}$, which is equal to 1 in case of unweighted graphs.

Node2vec[29] added search bias term $\alpha_{pq}$ to the transition probability which is based on parameters $p$, the return parameter, and $q$, the in-out parameter. The two parameters control the priority of the sampling strategy between breadth-first sampling and depth-first sampling.

**Edge-type transition matrix.** To deal with the different types of edges and their semantics when generating node sequences from heterogeneous networks, an edge-type transition matrix inspired from edge2vec[28] is used. An edge-type transition matrix is generated based on the correlations of edge-types consisting the network through an iterative Expectation-Maximization (EM) process. Given a heterogeneous network with $m$ types of edges, an edge-type transition matrix $M \in \mathbb{R}^{m \times m}$ is generated, where $M(i,j)$ refers to the transition weight between edge-types $i$ and $j$.

Initially, the walk paths are empty, and the transition matrix is initialized with all values set to 1. In the maximization step, a biased-random walk based on the transition matrix is performed to obtain $l$ walk paths. In the expectation step, the updated walk paths are used to optimize the transition matrix based on the correlation between edge-types. The updated values of the matrix are based on the sigmoid of the Pearson's correlation of the vectors of the edge-types, generated by counting the occurrence of each edge-type on each of the generated

paths. The updated value of the transition matrix can be expressed as:

$$M_{ij} = \text{Sigmoid}\left(\frac{E[(\mathbf{v}_i - \mu(\mathbf{v}_i))(\mathbf{v}_j - \mu(\mathbf{v}_j))]}{\sigma(\mathbf{v}_i)\sigma(\mathbf{v}_j)}\right)$$

where $\mathbf{v}_i, \mathbf{v}_j \in \mathbb{R}^l$ are vectors containing each edge-type's number of occurrence in the $l$ generated paths.

**Teleport operation and teleport factor.** Teleport operation is performed only when the type of current node $v$, $Type(v) \in$ {drug, disease}. Teleport factor $\tau$ where $0 \le \tau \le 1$, is a parameter that controls the rate of teleport operation and network traversing. In the proposed teleport-guided random walk algorithm, when the random walker arrives at a drug or disease node, the teleport action is chosen with a probability of $\tau$; otherwise, the network traversal continues, with a probability of $1 - \tau$.

For instance, if $\tau = 0.3$, the random walker on a drug/disease node has 30% probability of selecting teleport action and 70% probability of choosing network traversing. Thus, setting $\tau$ to a high value makes teleport operation more frequent, enabling the influence of semantic similarity to become greater and vice versa.

If the selected action of the random walker is teleport operation, the next node is randomly sampled from weighted probability as calculated in the similarity matrix $S$, defined through the process described in "Drug and disease semantic similarities". For example, given a current drug node $v_{drug}$, the probability for the next node to be drug node $n_{drug}$ through teleport operation can be express as below:

$$p(n_{drug}|\tau, v_{drug}, S) = \tau \times \frac{S_{v_{drug}, n_{drug}}}{\sum_{k_{drug} \in N_s(v_{drug})} S_{v_{drug}, k_{drug}}}$$

where $t$ is the teleport factor, and $N_s(v_{drug})$ is the neighboring node set of node $v_{drug}$ in the drug similarity matrix $S$. The same process is computed for disease nodes with disease similarity matrix $S^{disease}$.

**Network traversing.** Network traversing is performed when: 1) the current node type is other than drug/disease, or 2) the current node type is drug/disease and the selected action is network traversing. Given the current node $v$ and the previous node $u$ with the trained edge-type transition matrix $M$, the probability of selecting the next node $n$ can be shown as below:

$$p(n|v, u, M) = \sigma_{\tau}(v) \cdot \frac{w_{vn} \cdot M_{T(u,v)T(v,n)} \cdot \alpha_{pq}(n,u)}{\sum_{k \in N(v)} M_{T(u,v)T(v,k)} \cdot \alpha_{pq}(k,u)} \quad (3)$$

where $\sigma_{\tau}(v)$ is the network traverse probability defined by the node type of $v$, $N(v)$ is the neighboring node set of node $v$ and $T(u,v)$ is the edge-type between $u$ and $v$. Network traverse probability term is defined as follows:

$$\sigma_{\tau}(v) = \begin{cases} 1 - \tau & \text{if } Type(v) \in \{ \text{drug,disease} \} \\ 1 & \text{otherwise} \end{cases}.$$

## Node embedding generation

The Skip-gram[22] model is widely used for learning continuous feature representations of nodes in the random walk-generated node sequences (Multi-layered GBA through Teleport-guided randomwalk). The representation learning of the Skip-gram model is performed by optimizing a neighborhood-preserving likelihood objective function using stochastic gradient descent with negative sampling. However, the Skip-gram model does not consider the different types of nodes during the training steps, for example negative sampling step, thereby generating embedding vectors that are not aware of the node types. Inspired from an heterogeneous node representation learning model

metapath2vec++[78], we adopted a node type-aware heterogeneous Skip-gram approach for generating node embedding vectors from teleport-guided random walk-generated node sequences. By performing negative sampling only from nodes that belong in the same node type, the heterogeneous Skip-gram approach enables the generation of embedding vectors based on the distribution of node types within the graph.

During training, the `windowlength` parameter, that is the maximum distance between the current node and the predicted node in a node sequence, is set to 4 while training all models.

**Distance-based embedding space exploration.** "Embedding space of DREAMwalk exhibit the har-mony between biological and semantic information" explores the generated embedding spaces based on distance-based analyses. The distance between entities $u$ and $v$ are first measured through Euclidean distances using the equation:

$$d_{uv} = \sqrt{\sum_{i=1}^{n} (\mathbf{u}_i - \mathbf{v}_i)^2}$$

where $\mathbf{u}, \mathbf{v} \in \mathbb{R}^n$ refers to the embedding vectors of entities $u$ and $v$, respectively. While the Euclidean distance can be used for comparison within an embedding vector space, it is not suitable for comparing distances between two different spaces, such as embedding vector spaces created with different walk sequences. To enable the comparison of distances between teleport-guided and non-teleported spaces, we applied a normalization step to the calculated Euclidean distances of each space. This involved calculating the all-pairwise distance between every node in the network and then calculating the $z$-score normalized Euclidean distance for each pair. The normalization step involves subtracting the mean of each dimension and dividing by its standard deviation. The resulting $z$-score normalized Euclidean distance provides a standardized, unit-less measure of the dissimilarity between two data points.

To assess the statistical significance of the differences between $z$-score normalized distances of teleport-guided and non-teleported spaces, we conducted two-sided paired t-tests using the Scipy[79] Python package, which is prone to Python's underflow issues. To note, we observed cases where the resulting $P$ value was 0.0, indicating the occurrence of underflow. Underflow arises when numerical computation yields a value that is smaller than the minimum representable number for the data type being utilized, in this case, floating-data type, which can represent values between -$10^{-308}$ to $10^{308}$. Thus, any value smaller than -$10^{-308}$ was rounded to zero.

## XGBoost classifier for drug-disease association prediction
After the representations for all biomedical entities in the biomedKG are generated, an XGBoost[21] model is used for drug-disease association prediction and drug repurposing.

**XGBoost.** For learning the relationship between drug and disease, feature vectors obtained by element-wise subtraction of embedding vectors of two entities are used as input for an XGBoost classifier. XGBoost is a machine learning algorithm that is widely used in various application domains including computational drug discovery. Especially, XGBoost has proven to be highly effective for binary classification tasks with multiple features. XGBoost is an ensemble model that combines multiple weak learners, in this case decision trees, through a boosting algorithm to create a stronger learner. A boosting algorithm works by sequentially training weak learners based on the residual error of the previous learner. To prevent over-fitting during the training phase, number of boosting rounds were set to 500, and `maximumdepth` of below six.

**Drug-disease association prediction.** The DREAMwalk framework consists of two steps; node embedding generation step and link

prediction step. As mentioned above, we removed all the drug-disease links during the first step to generate drug and disease embedding with their MoA contexts. The drug-disease pairs were used in the second step; DDA prediction task as positive set. Negative drug-disease pairs of equal number of positive pairs were randomly sampled from the network. ten-times ten-fold cross-validation (CV) setting was adopted, and negative sampling was conducted in a way that there are no overlapping samples between the ten CV sets. The CV setting was applied identically to all the models evaluated for this study.

**Disease split for DDA prediction.** To assess the effectiveness of our model in a real-world drug repurposing scenario, we conducted an additional disease-split DDA prediction experiment on the MSI network. We first classified all disease entities based on their highest MeSH (Medical Subject Headings) term category, such as Cancers and other Neoplasms (Code: C04), Heart and Blood Vessel Diseases (C14). Next, we divided the categories into train, validation, and test sets using an approximate ratio of 8:1:1, repeating the process ten times. This method ensured that the drugs' efficacy in a new disease category was unknown during the training process, thus avoiding circularity.

**Window neighbor-based gene set enrichment analyses**
Gene set enrichment analysis is performed to analyze gene sets in various biological contexts. The node sequences generated through random walk algorithm opens the opportunity of interpreting of an entity's mechanism of actions. To narrow down the generated path to a gene set, we proposed an approach using the "window neighbors". Window neighbors of an entity is defined as the set of genes within a window of given size $l$ in the whole set of generated node sequences.

After retrieving the window neighbors with the window size of 4, enrichment analysis is performed to identify the associated functional terms using Enrichr[42]. During the enrichment analysis, two statistical tests are applied. First, to determine whether there is a significant association between the window neighbor set and a given KEGG pathway or Molecular Function, Fisher's exact test is utilized. Another statistical method utilized is Benjamini-Hochberg correction. This false discovery rate adjustment method is used to adjust for the multiple hypothesis testing involved in analyzing large datasets. The resulting Benjamini-Hochberg correction yields the final adjusted $P$ value to identify biologically meaningful associations between the window neighbor sets and various functional contexts. Also, the gene ratio represents the number of overlapping genes divided by the number of genes in assigned to the given term, and interaction size refers to the number of the overlapping genes.

**Statistics and reproducibility**
For all performance comparison of DREAMwalk's teleport-guided embedding space with non-teleported embedding space, we have performed two-sided paired t-tests for evaluating the significance between the two spaces. No statistical method was used to pre-determine sample size, and no data were excluded from the analyses.

**Reporting summary**
Further information on research design is available in the Nature Portfolio Reporting Summary linked to this article.

## Data availability
Each biomedKGs were retrieved from the corresponding GitHub repositories and through API calls; HetioNet (https://github.com/hetio/hetionet), MSI (https://github.com/snap-stanford/multiscale-interactome) KEGG (https://www.kegg.jp/kegg/rest/keggapi.html). Source data for all Figures are provided with this paper in the Source Data file. Source data are provided with this paper.

## Code availability
The source code for DREAMwalk's node embedding and DDA prediction are available at the following GitHub repository (https://github.com/eugenebang/DREAMwalk) under the DOI: 10.5281/zenodo.7935342)[80].

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

## Acknowledgements

This research was supported by the Bio & Medical Technology Development Program of the National Research Foundation (NRF), funded by the Ministry of Science & ICT(NRF-2022M3E5F3085677, NRF-2022M3E5F3085681, RS-2023-00257479). The ICT at Seoul National University provided research facilities for this study.

## Author contributions

D.B., S. Lim, S. Lee, and S.K. conceived the experiments, D.B. conducted the experiments, D.B., S. Lim, S. Lee, and S.K. analyzed the results, D.B. wrote the manuscript. All authors reviewed the manuscript.

## Competing interests

The authors declare no competing interests.
