## [Peer Review File · Nature Communications]

Biomedical knowledge graph learning for drug repurposing by extending guilt-by-association to multiple layersReviewer #1 (Remarks to the Author):

Bang et al. describe DREAMwalk, an approach that uses random walks on a multi-layer drug-disease-gene network to identify drug repurposing opportunities. DREAMwalk considers random walks with teleportation (an established algorithm for analyzing social, information, and biological networks). Each time a random walker makes a step by following edges in the network, it can (with a small probability) teleport itself to a node in the network that is not connected to the current node by an edge. The teleportation process is biased, so it preferentially selects biologically similar drugs and diseases. The basis of this approach is that similar drugs share disease-relevant therapeutic targets.

After the random walk process is completed, Bang et al. use a skip-gram model (a well-known embedding technique introduced in 2013 as a word2vec algorithm and in 2015 as a node2vec algorithm) to create embedding vectors which are used for predicting drug-disease associations as drug repurposing.

The paper is well-written and easy to follow. Case studies in section 2.3 are insightful -- they consider the biological mode of action of drugs (MoA) and their relationship with diseases to illustrate how DREAMwalk can distinguish drugs with different MoAs but the same therapeutic targets.

However, I have several major concerns with this paper, including limited methodological novelty, incomplete performance evaluation, lacking external/independent evaluation, and no clinical evaluation, which are detailed below. Unfortunately, I cannot recommend this paper for publication.

1) DREAMwalk is advertised as a "clinical information-guided random walk." Bang et al. extensively write about integrating "clinical information" into their model. Unfortunately, the paper does NOT consider any clinical information for model development, training, or evaluation. NO patient information is considered. NO clinical/healthcare/EHR or EHR-derived information is used.

As evidenced by section 4.2 (page 13), "clinical disease-disease similarities" and "clinical drug-drug" similarities are semantic similarity measures calculated using non-clinical ontologies. Bang et al. use ATC classification (<https://www.who.int/tools/atc-ddd-toolkit/atc-classification>) for drugs and the medical vocabulary of disease concepts (<https://disease-ontology.org/> + MeSH) for diseases. These data resources provide standardized ontologies for drugs and diseases and disease. They are considered in many other drug repurposing papers (e.g., <https://www.ncbi.nlm.nih.gov/pmc/articles/PMC9294412/>, <https://academic.oup.com/bioinformatics/article/32/17/2664/2450730>, <https://academic.oup.com/bioinformatics/article/38/5/1369/6454947>). However, these are not clinical resources. Therefore, it is incorrect to refer to them as "clinical information" throughout the paper.

2) Limited novelty of the algorithmic approach. The overall approach has two components, first, biased random walks with teleportation, and second, shallow skip-gram network embeddings. Individually, these two elements have been used extensively in network biology in the last decade. Taken together, these two elements have been described in several drug repurposing papers.

Following is a non-exhaustive list of a handful of examples:

- <https://www.ncbi.nlm.nih.gov/pmc/articles/PMC8459562/>
- <https://ieeexplore.ieee.org/stamp/stamp.jsp?arnumber=9176165>
- <https://bmcbioinformatics.biomedcentral.com/articles/10.1186/s12859-019-3117-6>
- <https://academic.oup.com/bioinformatics/article/36/4/1241/5581350>
- <https://arxiv.org/pdf/2004.14842.pdf>
- <https://www.sciencedirect.com/science/article/pii/S1532046421001672>

3) To benchmark DREAMwalk, Bang et al. consider random drug-disease split based on approved indications and temporal split based on drugs' approval date. Both these splits, however, can be limited due to circularity. The authors should consider splits based on disease areas or therapeutic

areas, where all diseases in a given area (such as cancers or cardiovascular diseases) are in the test set, and no disease from the area is in the training set. Otherwise, the model can find a short-cut proxy disease in the training set similar to the test disease (clinically similar, even though these might be two different disease nodes in the network) and copy its indications to the test disease. This can lead to circular prediction and unrealistic estimates of performance.

4) Statistical analyses.

- In Figure 3e, what are the distances between other hypertensive drug pairs?
- In Table 1, only top-10 predictions are shown. Can you include a negative control experiment and investigate the bottom of prediction lists? Similarly, where in the ranked list do we find known drug-disease indications? Can predictions be confounded by indication (e.g., due to comorbidities)?
- Bang et al. write that the top 10 candidate drugs for breast carcinoma include chemotherapeutic agents that are often used off-label for metastatic breast cancers and other metastatic cancers in the clinic. Interesting observation; however, it suggests data leakage. This is because these diseases are represented by different disease concepts/notes in the dataset (network). However, clinically, they are therapeutically related. Therefore, bang et al. should consider including off-label use information in the dataset and use it in the train-test split.

Reviewer #2 (Remarks to the Author):

This manuscript proposed a computational framework DREAMwalk to predict drug-disease associations and drug repurposing by using the clinical neighbors of drug and disease entities and a semantic multi layer guilt-by-association. Generally speaking, this paper is well written, and the whole problem solving protocol was clearly described. The authors should address the following problems:

- (1) The authors used the skip-gram model for learning continuous feature representations of nodes, the reason should be further explained why this method but not other embedding model is selected in the manuscript. Did the authors try other methods for the same aim?
- (2) There are some small mistakes. Thus, please make sure that it has been proofread to avoid possible grammar and typology errors. For example, in line 76 'Based on these ideas, we propose DREAMwalk: Drug Repurposing through Exploring Associations using Multi-layer random walk' .
- (3) The author omitted some citations in the manuscript.
- (4) The methodology section is a little bit simple.
- (5) The authors fused various information, is there any noisy information among these different data?
- (6) Why was ten fold cross-validation chosen instead of the usual five fold?

Reviewer #3 (Remarks to the Author):

KEY RESULTS

This paper describes a new method, DREAMwalk, for performing node embedding on biomedical knowledge types. It adapts existing and established random walk techniques by randomly "teleporting" to nodes of similar types when performing random walks. The authors use their algorithm to propose (1) potential mechanisms of action by which existing drugs operate in the body and (2) potential drug repurposing candidates for Alzheimer's disease and breast cancer.

VALIDITY

The key conclusions of the paper are as follows:

DREAMwalk embeds drugs with similar use closer together than baselines. This seems to be supported qualitatively by Figure 3 but is not quantitatively substantiated in general.

DREAMwalk identifies drug mechanisms of action better than existing methods. This given data seem insufficient to properly support this conclusion. Most of the results related to MoA prediction are summarized in Figure 5. In this figure, mechanism of action is identified by plugging a group of "window neighbor" proteins (nodes within k steps of the drug in a random walk) into a third party grouping tool, which identifies mechanisms and statistical significance. Mechanisms were sorted in descending order relative to paths produced by DREAMwalk. However, the authors fail to show mechanisms sorted in descending order of the baselines, which does not allow for an unbiased comparison. This makes it difficult to determine if DREAMwalk is truly better than baselines.

DREAMwalk identifies novel & meaningful drug repurposing candidates better than baselines. The given data are not sufficient to support this conclusion. First, Table 1 omits a number of highly ranked drug repurposing candidates. All of the top ranked candidates should be included here because it allows the reader to better evaluate potential failure modes of the algorithm. It would additionally be helpful to compare the top candidates produced by DREAMwalk to those by another algorithm, e.g. residual2vec to see how they compare and if DREAMwalk's candidates are demonstrably better.

SIGNIFICANCE

The proposed model makes an incremental improvement on existing graph embedding methodology. Most of the baselines compared with are relatively old and outdated -- more recent models such as SEAL show better link prediction performance on public benchmarks. Additionally, this paper does not consider more general knowledge graph link prediction methods (e.g. ComplEx, RotatE, QuatE, etc) to see if this class of model may have better drug-disease association (DDA) performance than methods investigated.

It is worth noting that the vast reported repurposing results in Table 1 seem to identify drugs used for very similar diseases (e.g. other cancer treatments for breast cancer; PD treatments for AD). Without providing any associated context for these predictions, the result indicate that DREAMwalk struggles to identify truly novel drug repurposing candidates beyond what could be deduced via visual inspection by a clinician.

DATA AND METHODOLOGY

Performance of MoA prediction needs more data to properly validate (see second bullet of "Validity" section). Novelty of drug repurposing candidates needs more evidence to conclusively demonstrate if DREAMwalk is better than existing methods (see "Validity"). Drug repurposing candidates are omitted from Table 1 making it impossible to determine if best candidates were "cherry picked" or if they are truly representative of all results.

ANALYTICAL APPROACH

Paper needs better details of what statistical tests were used to compare score distributions and determine significance.

Suggested Improvements include:

Show mechanisms of action and drug repurposing candidates identified by baseline algorithms to show if/where DREAMwalk may be worse than existing methods

Show all top-ranked candidates

Use more (and more recent) baselines to compare DDA prediction performance (e.g. KG embedding methods, top methods from link prediction section of Open Graph Benchmark)

CLARITY AND CONTEXT

This paper needs to include additional graph mining algorithms as performance baselines in

identifying drug repurposing candidates. This would give better context and clarity for if and how exactly DREAMwalk's results differ from existing results.

Response to reviewer's comments

We would like to thank all the reviewers for the insightful and helpful suggestions. According to the comments, we tried to improve the quality of the manuscript through additional experiments and by further discussions on model performances and findings. Please see below our detailed response to reviewers' comments in blue.

Reviewer #1 (Remarks to the Author):

Bang et al. describe DREAMwalk, an approach that uses random walks on a multi-layer drug-disease-gene network to identify drug repurposing opportunities. DREAMwalk considers random walks with teleportation (an established algorithm for analyzing social, information, and biological networks). Each time a random walker makes a step by following edges in the network, it can (with a small probability) teleport itself to a node in the network that is not connected to the current node by an edge. The teleportation process is biased, so it preferentially selects biologically similar drugs and diseases. The basis of this approach is that similar drugs share disease-relevant therapeutic targets.

After the random walk process is completed, Bang et al. use a skip-gram model (a well-known embedding technique introduced in 2013 as a word2vec algorithm and in 2015 as a node2vec algorithm) to create embedding vectors which are used for predicting drug-disease associations as drug repurposing.

The paper is well-written and easy to follow. Case studies in section 2.3 are insightful -- they consider the biological mode of action of drugs (MoA) and their relationship with diseases to illustrate how DREAMwalk can distinguish drugs with different MoAs but the same therapeutic targets.

Thank you very much for the detailed summary and valuable comments.

However, I have several major concerns with this paper, including limited methodological novelty, incomplete performance evaluation, lacking external/independent evaluation, and no clinical evaluation, which are detailed below. Unfortunately, I cannot recommend this paper for publication.

1) DREAMwalk is advertised as a "clinical information-guided random walk." Bang et al. extensively write about integrating "clinical information" into their model. Unfortunately, the paper does NOT consider any clinical information for model development, training, or evaluation. NO patient information is considered. NO clinical/healthcare/EHR or EHR-derived information is used.

As evidenced by section 4.2 (page 13), "clinical disease-disease similarities" and "clinical drug-drug" similarities are semantic similarity measures calculated using non-clinical ontologies. Bang et al. use ATC classification (<https://www.who.int/tools/atc-ddd-toolkit/atc-classification>) for drugs and the medical vocabulary of disease concepts (<https://disease-ontology.org/> + MeSH) for diseases. These data resources provide standardized ontologies for drugs and diseases and disease. They are considered in many other drug repurposing papers (e.g.

<https://www.ncbi.nlm.nih.gov/pmc/articles/PMC9294412/>,

<https://academic.oup.com/bioinformatics/article/32/17/2664/2450730>,

<https://academic.oup.com/bioinformatics/article/38/5/1369/6454947>). However, these are not clinical resources. Therefore, it is incorrect to refer to them as "clinical information" throughout the paper.

Figure R1-1. The drug-disease association prediction performances of three similarity-utilizing models on the three biomedKGs. a,b,c DDA prediction performance on MSI network, HetioNet and KEGG network with random split, respectively. t-test rel p: resulting p-value of paired t-test

We thank the reviewers for the valuable comment. We agree that our method utilizes *semantic* similarities, but not *clinical* information as we did not use clinical information/healthcare/EHR-derived information. Thank you very much for pointing out the critical mistake. After careful consideration of your feedback, we have revised our terminology from '*clinical* level information' and '*clinical* similarity' to '*semantic* level information' and '*semantic* similarities', respectively.

2) Limited novelty of the algorithmic approach. The overall approach has two components, first, biased random walks with teleportation, and second, shallow skip-gram network embeddings. Individually, these two elements have been used extensively in network biology in the last decade. Taken together, these two elements have been described in several drug repurposing papers.

Following is a non-exhaustive list of a handful of examples:

- <https://www.ncbi.nlm.nih.gov/pmc/articles/PMC8459562>

Drug-disease association prediction performance on MSI

Figure R1-2. The drug-disease association prediction performances using four different embedding methods on MSI with random splits. Prediction performance of Heterogeneous SG is significantly higher than that of SG, CBOW and end-to-end neural network embedding methods. SG: Skip-gram; t-test rel p : resulting p -value of paired t-test.

- <https://ieeexplore.ieee.org/stamp/stamp.jsp?arnumber=9176165>
- <https://bmcbioinformatics.biomedcentral.com/articles/10.1186/s12859-019-3117-6>
- <https://academic.oup.com/bioinformatics/article/36/4/1241/5581350>
- <https://arxiv.org/pdf/2004.14842.pdf>
- <https://www.sciencedirect.com/science/article/pii/S153204642100167>

We appreciate very much for your kindness to let us know the state-of-the-art technologies. We agree that biased random walk and shallow skip-gram network embedding-based models have been established a long time ago.

We would like to highlight that the novelty of our work lies in generating an “embedding space for drug and disease” through genes by performing random walks with the ‘multi-layer Guilt-by-association (GBA)’-based technology. The ultimate goal of drug repurposing is to identify relations between existing drugs and known diseases. To achieve this goal, we created embedding space of both drug and disease by populating paths connecting drugs and diseases through genes. The unique contribution of our work lies in the way we generate this single embedding space of drug and disease.

We would like to emphasize that the gene network part of the knowledge graph is much denser than the drug and disease networks, making it a major hurdle to connect drug and disease using current biased-random walk approaches. We overcome this challenge by introducing semantic neighbors of drugs and diseases, which enables the creation of many paths passing through drug and disease, resulting in an effective embedding space which enables accurate prediction of drug-disease associations.

In response to your suggestion, we explored and examined existing embedding methods and random walk generation methods including methods suggested by reviewers: DTi2vec (Thafar et al., 2019) and NEWMIN (Yu et al., 2022), as baseline models (**Figure R1-1**). We would like to clarify that our major changes in computational methods involved the use of heterogeneous skip-gram and XGBoost classifier, which led to a 3.73% improvement in accuracy.

Drug-disease association prediction performance on MSI with disease split

Figure R1-3. The drug-disease association prediction performances of comparison methods on MSI with disease splits. t-test rel p: resulting p-value of paired t-test.

Our experiments show that DREAMwalk achieved the highest scores for accuracy, AUROC, and AUPR in predicting drug-disease associations (DDAs) of the three biomedical knowledge graphs (biomedKGs), with a statistically significant difference of p -value below 0.05 through pair-wise t-test compared to DTi2vec and NEWMIN. Our results demonstrate that an approach based on learning molecular-level KG with semantic information as its guide results in improved prediction of drug-disease association.

Also, as mentioned, we improved our model by considering the heterogeneity of the biomedKGs in the embedding step through a heterogeneous Skip-gram algorithm inspired from Metapath2vec++ (Dong et al., 2013). Heterogenous Skip-gram differs from conventional Skip-gram by considering node types when performing negative sampling for node embedding generation. To the best of our knowledge, DREAMwalk is the first model to adopt a heterogeneous shallow embedding approach for drug-disease association prediction and drug repurposing, and we demonstrate that the application of the heterogeneous Skip-gram resulted in improved DDA prediction performance compared to homogeneous embedding algorithms, including Continuous Bag-of-words (CBOW) and end-to-end neural network (Figure R1-2).

We have included these results in the Results and Supplementary Figure of the revised manuscript to address the reviewers' comments.

3) To benchmark DREAMwalk, Bang et al. consider random drug-disease split based on approved indications and temporal split based on drugs' approval date. Both these splits, however, can be limited due to circularity. The authors should consider splits based on disease areas or therapeutic areas, where all diseases in a given area (such as cancers or cardiovascular diseases) are in the test set, and no disease from the area is in the training set. Otherwise, the model can find a short-cut proxy disease in the training set similar to the test disease (clinically similar, even though these might be two different disease nodes in the network) and copy its indications to the test disease. This can lead to circular prediction and unrealistic estimates of performance.

Figure R1-4. The all-pairwise normalized Euclidean distance between treatments with shared indications. Due to the underflow issue of python, any value of below $10E-308$ is represented as 0.0. t-test rel p: resulting p-value of paired t-test.

We would like to thank the reviewer for this thoughtful comment. We completely agree that both random-split and temporal-split benchmark experiments carry the limitations due to circularity. To address this issue, we conducted an additional disease-split DDA prediction experiment on the MSI network to evaluate the performance of our model in a more representative real-world drug repurposing setting.

We categorized all disease entities in the highest MeSH (Medical Subject Headings) term category, such as Cancers and other Neoplasms (C04), Heart and Blood Vessel Diseases (C14). Then, we assigned data into train, validation, test set with approximately 8:1:1 ratio for 10 times. This process ensures that the drugs' function in a brand-new category of disease is unknown during the training process, thus preventing circularity.

In this setting, the prediction problem became more challenging, all models resulted in decrease in prediction performance of up to 40% (in the case of QuatE (Zhang et al., 2019)) (Figure R1-3). However, we observed that DREAMwalk achieves the highest accuracy in terms of AUROC and AUPRC, compared to baseline methods in the disease-split case. We believe that this is possible since the representation of the entities, especially protein nodes, learn not only the biological context but also the *semantic* information propagated from neighboring drugs/diseases. We thank the reviewer for this comment and added the corresponding new results to Results section of the revised manuscript which demonstrates our framework's ability in more real-world likely settings.

4) Statistical analyses.

- In Figure 3e, what are the distances between other hypertensive drug pairs?

Thank you for bringing this to our attention. Although we have displayed the decrease of the normalized Euclidean distances among the three anti-hypertensive drugs in Figure 3e of the previous version of the manuscript, we lacked a quantitative demonstration of the decrease in distances between drugs sharing the same indication through teleport-guided random walk.

Table R1-1. Top-10 and bottom-10 list of the drug repurposing candidates by DREAMwalk for Breast carcinoma and Alzheimer's disease. Avg. Prob: average probability; SD : Standard deviation

Breast Carcinoma				
Rank	Drug	Original indication	Avg. Prob.	SD
1	Irinotecan	Colorectal cancer, SCLC, NSCLC	0.9949	0.00332
2	Etoposide	Germ cell tumors, Kaposi sarcoma, SCLC	0.9930	0.00662
3	Dactinomycin	Wilm's tumor, Rhabdomyosarcoma, Neuroblastoma	0.9916	0.01278
4	Teniposide	ALL, Small Cell Carcinoma, Lymphoid Leukemia	0.9906	0.01091
5	Vinblastine	Hodgkin disease, Lymphoma, NHL	0.9890	0.01246
6	Mitoxantrone	AML, Multiple Sclerosis, Lymphoma, Sarcoma	0.9888	0.01279
7	Interferon alfa-2b	Melanoma, Brain Neoplasms, Hepatitis C	0.9884	0.00954
8	Cortisone-acetate	SLE, CTCL, IBD, Autoimmune Diseases	0.9871	0.02182
9	Vindesine	CML, Melanoma, ALL, Hodgkin's lymphoma	0.9870	0.01003
10	Hydroxyurea	CML, cancer of head and neck, sickle cell anemia	0.9862	0.01279
...
1641	lacosamide	Epilepsy, seizures	0.0012	0.00079
1642	phenacemide	Epilepsy, Epilepsy, Complex Partial, seizures	0.0012	0.00087
1643	ajmaline	Cardiac Arrhythmia	0.0011	0.00118
1644	enoximone	congestive heart failure	0.0011	0.00086
1645	molsidomine	coronary artery disease (CAD)	0.0011	0.00090
1646	Encainide	Premature Ventricular Contractions, Cardiac Arrhythmia	0.0010	0.00097
1647	tocainide	ventricular arrhythmias, Cardiac Arrhythmia	0.0010	0.00078
1648	mephenytoin	seizures	0.0009	0.00058
1649	moracizine	ventricular arrhythmias, Cardiac Arrhythmia	0.0008	0.00095
1650	vernakalant	Cardiac Arrhythmia, atrial fibrillation (AF)	0.0006	0.00066
Alzheimer's disease				
Rank	Drug	Original indication	Avg. Prob.	SD
1	Levetiracetam	Simple partial seizures, Epilepsy	0.9901	0.01375
2	clomipramine	OCD, Chronic pain, Narcolepsy	0.9825	0.01614
3	duloxetine	Major depressive disorder, Peripheral neuropathy	0.9767	0.02362
4	fluoxetine	Major depressive disorder, OCD, Bipolar disorder	0.9755	0.0315
5	maprotiline	Depressive disorder, Duodenal Ulcer	0.9743	0.02527
6	armodafinil	Narcolepsy, Obstructive sleep apnea	0.9737	0.02783
7	sertraline	Depressive disorder, OCD, Panic disorders	0.9731	0.03506
8	Lisdexamfetamine	ADHD	0.9711	0.03051
9	atomoxetine	ADHD	0.968	0.05088
10	dextroamphetamine	ADHD, narcolepsy	0.9666	0.03925
...
1641	Enoxacin	Urinary tract infection, Cystitis	0.0062	0.00525
1642	itraconazole	onychomycosis	0.0061	0.00582
1643	benralizumab	asthma	0.006	0.00747
1644	penicillin-v-potassium	Erysipelas, Gingivostomatitis, pneumonia	0.0059	0.00389
1645	ofatumumab	chronic lymphocytic leukemia	0.0058	0.00737
1646	lomefloxacin	Urinary tract infection, Lower respiratory tract infection, tuberculosis	0.0053	0.00416
1647	sparfloxacin	Pneumonia, tuberculosis	0.0044	0.00474
1648	pyrazinamide	tuberculosis	0.0039	0.00314
1649	trovafloxacin	Urinary tract infection, sinusitis	0.0039	0.0034
1650	fluconazole	meningitis	0.0032	0.0029

To address this issue, we conducted additional analyses to calculate the pairwise distances between all anti-hypertensive drugs (a total of 84) in the MSI network. The results demonstrate that the drugs are indeed located closer to each other in the teleport-guided embedding space, compared to the ones generated without teleportation, which further supports our previous findings (**Figure R1-4**). We also performed similar analyses for drugs used in the treatments of rheumatoid arthritis, asthma, and allergic rhinitis, and we found that results were consistent with our previous findings. These additional analyses further validate the effectiveness of our teleport-guided approach for generating drug embeddings that capture semantic information and allow for identification of drugs with common indications. We included these results in the Results section of the revised manuscript.

Table R1-2. Top-10 drug repurposing candidates for Alzheimer’s disease and breast carcinoma of baseline models.

Alzheimer’s disease						
Rank	DTI2vec	NEWMIN	WalkPool	SEAL	ComplEx	edge2vec
1	promazine	clonazepam	isoflurophate	Zinc	dopamine	dexamethasone
2	melatonin	carbamazepine	mestion	Cyclophosphamide	amantadine	hydrocortisone
3	felbamate	propentofylline	Choline	Ritonavir	Phenobarbital	prednisolone
4	tetrabenazine	sertraline	choline salicylate	Rasagiline	norepinephrine	triamcinolone
5	gabapentin	topiramate	dipivefrine	Melatonin	ifosfamide	Interferon alfa-2b
6	diazepam	tiagabine	Mecasermin	Resveratrol	leuprolide	fludrocycortide
7	lorazepam	levetiracetam	edrophonium	Mecasermin	Theophylline	prednisone
8	phenytoin	ramelteon	pralidoxime	Tranylcypromine	cisplatin	rimexolone
9	clonazepam	retigabine	pralidoxime-chloride	Diacerein	baclofen	methylprednisolone
10	Methamphetamine	Chlordiazepoxide	demecarium	Choline salicylate	orphenadrine	buphenine
Breast Carcinoma						
Rank	DTI2vec	NEWMIN	WalkPool	SEAL	ComplEx	edge2vec
1	Canakinumab	Interferon alfa-2b	Thalidomide	Succinic acid	etoposide	dexamethasone
2	methotrexate	vinblastine	fostamatinib	Aspirin	vincristine	hydrocortisone
3	Interferon alfa-2b	Peginterferon alfa-2a	dexrazoxane	Arsenic trioxide	hydroxyurea	triamcinolone
4	vinblastine	carmustine	teniposide	dexamethasone	irinotecan	prednisolone
5	Thalidomide	Interferon Alfa-2a	etoposide	arsenic-trioxide	cisplatin	prednisone
6	thiotepa	vincristine	daunorubicin	prednisolone	dactinomycin	Interferon alfa-2b
7	methylprednisolone	Interferon alfacon-1	clofarabine	hydrocortisone	vinblastine	fludrocycortide
8	cisplatin	hydroxyurea	dexamethasone	amcinonide	bleomycin	buphenine
9	Interferon beta-1a	Interferon beta-1a	dactinomycin	urea	carmustine	methylprednisolone
10	azathioprine	Peginterferon alfa-2b	cytarabine	prednisone	ifosfamide	Interferon beta-1b

Table R1-3. DREAMwalk’s predicted probabilities of known treatments for Breast Carcinoma and Alzheimer’s Disease. Avg. Prob: average probability; SD : Standard deviation

Rank	Breast Carcinoma			Alzheimer’s Disease		
	Drug	Avg. Prob.	SD	Drug	Avg. Prob.	SD
1	doxorubicin	0.9950	0.00533	citicoline	0.9927	0.00378
2	docetaxel	0.9933	0.00444	memantine	0.9920	0.00664
3	epirubicin	0.9918	0.00531	rivastigmine	0.9888	0.00541
4	5-fluorouracil	0.9912	0.00604	donepezil	0.9858	0.01671
5	cyclophosphamide	0.9912	0.01313	NADH	0.9802	0.01182
6	paclitaxel	0.9885	0.01925	tacrine	0.9697	0.04921
7	ixabepilone	0.9884	0.01357	galantamine	0.9638	0.04912
8	everolimus	0.9796	0.00836	Valproic Acid	0.8667	0.2701
9	Nandrolone decanoate	0.9712	0.03032	Vitamin E	0.8641	0.29124
10	megestrol acetate	0.9592	0.04337	selegiline	0.8487	0.25281
11	Nandrolone phenpropionate	0.9260	0.14429	physostigmine	0.7528	0.35803

Table R1-4. Predicted probabilities of each model on actual drug repurposing cases. The highest probability for each drug-disease pair is highlighted in bold. ED: Erectile dysfunction; MI: Myocardial infarction; MM: Multiple Myeloma; PAH: Pulmonary arterial hypertension; RCC: Renal Cell Carcinoma

Repurposing case	DREAMwalk	NEWMIN	DTi2vec	SEAL	WalkPool	node2vec	edge2vec	CompIEx
Aspirin-MI	0.9987	0.9946	0.9949	0.9171	0.9504	0.7559	0.567	0.692
Aspirin-Thrombosis	0.9864	0.9844	0.9695	0.8746	0.8276	0.7229	0.6923	0.4608
Sildenafil-ED	0.9523	0.6174	0.4708	0.6958	0.632	0.4309	0.6299	0.3345
Sildenafil-PAH	0.3554	0.5247	0.4764	0.4202	0.4382	0.4170	0.6695	0.1665
Thalidomide-MM	0.9772	0.9964	0.9896	0.9129	0.6886	0.5986	0.4951	0.6499
Thalidomide-RCC	0.6023	0.8847	0.9632	0.8237	0.6539	0.6569	0.4659	0.4896
Finasteride-Male Alopecia	0.4087	0.2413	0.2824	0.0927	0.2326	0.3179	0.3072	0.172
Minoxidil-Alopecia	0.1478	0.1451	0.2398	0.1203	0.161	0.3740	0.3205	0.3427
Median	0.7773	0.7511	0.7198	0.7598	0.6430	0.5148	0.5311	0.4018

- In Table 1, only top-10 predictions are shown. Can you include a negative control experiment and investigate the bottom of prediction lists? Similarly, where in the ranked list do we find known drug-disease indications? Can predictions be confounded by indication (e.g., due to comorbidities)?

We thank the reviewer for their valuable suggestion. Responding to the first question, we listed the complete top-10 and lowest-10 predictions of our model in **Table R1-1**. Upon examining the lowest-10 drug lists, we observed that drugs from therapeutically unrelated areas, such as antibiotics for Alzheimer’s disease, and cardiac arrhythmia and epilepsy drugs for breast cancer, were present in the list. It is important to note that our model was updated from using a Multi-layer perceptron (MLP) classifier to an eXtreme Gradient Boosting (XGBoost) classifier, which resulted in a different drug top-10 list from the previously submitted version. Additionally, we provided the top-10 list generated by baseline models, namely DTi2vec, NEWMIN, WalkPool, SEAL, CompIEx, and edge2vec (**Tables R1-2**). By comparing the lists, we conclude that our model found repurposing candidates from diverse therapeutic areas that are well supported in the literature.

Table R1-5. Predicted probabilities and rankings of repurposing candidate drugs in phase 3 clinical trial for Alzheimer’s disease in 2021. For each model, the drug-disease association probabilities were measured for 10 times, then ranked by their average probabilities. The ranks of the drugs are shown in parentheses, and the highest probability and rank for each drug are highlighted in bold.

Drugs	DREAMwalk	edge2vec	ComplEx	DTi2vec	node2vec	NEWMIN	WalkPool	SEAL
Brexpiprazole	0.736 (210)	0.499 (1496)	0.466 (886)	0.769 (211)	0.338 (1631)	0.692 (120)	0.237 (1471)	0.338 (1085)
Caffeine	0.910 (59)	0.582 (975)	0.680 (49)	0.706 (238)	0.556 (204)	0.576 (273)	0.620 (179)	0.518 (247)
Escitalopram	0.875 (82)	0.616 (738)	0.412 (1180)	0.870 (104)	0.439 (982)	0.596 (248)	0.227 (1538)	0.426 (699)
Guanfacine	0.602 (326)	0.584 (962)	0.503 (698)	0.110 (970)	0.477 (606)	0.431 (617)	0.239 (1458)	0.281 (1396)
Hydralzaine	0.481 (412)	0.626 (665)	0.678 (51)	0.110 (968)	0.508 (366)	0.391 (925)	0.436 (514)	0.337 (1086)
Metformin	0.030 (1440)	0.531 (1362)	0.531 (549)	0.047 (1346)	0.350 (1618)	0.38 (1037)	0.386 (747)	0.341 (1071)
Omega-3-carboxylic acids	0.342 (564)	0.592 (912)	0.409 (1199)	0.443 (456)	0.404 (1327)	0.436 (589)	0.509 (292)	0.363 (980)
Median	0.602 (326)	0.584 (962)	0.503 (698)	0.443 (456)	0.439 (982)	0.436 (589)	0.386 (747)	0.341 (1071)

For example, on the repurposing lists for breast carcinoma with NEWMIN or edge2vec's list, we observed that while NEWMIN and edge2vec are not successful in finding drug candidates other than interferon derivatives and corticosteroids (drugs with suffix '~sone/lone'). In contrast, DREAMwalk ranked highly not only chemotherapeutic agents but also interferon derivatives and corticosteroids. These results demonstrate the ability of our model to identify novel and meaningful drug repurposing candidates.

Regarding the second question, we chose not to include known drug-disease indications in our list because our main objective was to discover potential treatment options from unknown drug-disease associations. Additionally, since the known drug-disease associations are mostly located in the training set, the classifier would have already learned those associations during the training process, which makes the relations unsuitable for inference. However, we did provide the list of predicted probabilities of the known drug-disease associations for breast carcinoma and Alzheimer’s disease in **Table R1-3**, which demonstrates our model's ability to predict known treatments with high accuracy, with 18 out of 22 associations predicted over 0.9.

In response to the reviewer's concern about confounding factors such as comorbidities, we acknowledge that this is an important consideration. This is indeed a very important and critical suggestion. However, to address this issue, we would need access to patient-specific networks, which are currently unavailable in the public domain. We are excited to investigate this important question in the future, but this topic is currently out of the scope of our study.

- Bang et al. write that the top 10 candidate drugs for breast carcinoma include chemotherapeutic agents that are often used off-label for metastatic breast cancers and other metastatic cancers in the clinic. Interesting observation; however, it suggests data leakage. This is because these diseases are represented by different disease concepts/notes in the dataset (network). However, clinically, they are therapeutically related. Therefore, bang et al. should consider including off-label use information in the dataset and use it in the train-test split.

Thank you for the thoughtful comment. We appreciate the reviewer's concerns regarding the

representation of cancer types in the biomedical Knowledge Graphs and agree that cancers are therapeutically related, despite being mostly represented by their anatomy (breast cancer, renal cell, hepatocyte, etc.). We agree that some diseases are therapeutically related clinically, e.g., metastatic cancers. However, we would point out that cancers are known as many diseases and relations among anatomically different cancers are yet to be well understood. Thus, it is our opinion that prediction over anatomically different cancers is not likely due to the information leakage. As we have better knowledge on the core mechanisms of anatomically different cancers in the future, we may provide better response to your thoughtful comment, but we believe that our results are not from the information leakage.

Nonetheless, we highly value the reviewer's comment and have performed two additional experiments to address the concerns. First, we conducted an actual drug repurposing split experiment using eight well-known drug repurposing cases in the test set (Jourdan et al., 2020). Our model achieved the highest mean probability over all models, outperforming baseline models in four out of eight cases, along with four cases having a high confidence level with an average probability of over 90% (**Table R1-4**). Second, we evaluated the performance of our model in the clinical setting by predicting the drug-disease associations (DDAs) of Alzheimer's disease (AD) repurposing candidates in phase 3 clinical trials as of 2021 (Cummings et al., 2022). We focused on eight drugs: Brexpiprazole, Caffeine, Escitalopram, Guanfacine, Hydralazine, Metformin, and Omega-3-carboxylic acids and their association with AD. Our model predicted the DDAs with the highest median probability and rank compared to the seven baseline models (**Table R1-5**). Notably, two drugs (Caffeine and Escitalopram) had a predicted probability of over 0.8, indicating a strong likelihood of a drug-disease link that no other models were able to output except for DTi2vec.

We believe that these additional experiments, that evaluated our model's performance on actual drug repurposing cases and phase 3 clinical trials of AD, respectively, provide compelling evidence of the effectiveness of our model without data leakage concerns. We have included the results of these experiments in Supplementary Table 7 and the Results section of the revised manuscript, respectively, as per the reviewer's suggestion.

Reviewer #2 (Remarks to the Author):

This manuscript proposed a computational framework DREAMwalk to predict drug-disease associations and drug repurposing by using the clinical neighbors of drug and disease entities and a semantic multi layer graph-by-association. Generally speaking, this paper is well written, and the whole problem solving protocol was clearly described. The authors should address the following problems:

(1)The authors used the skip-gram model for learning continuous feature representations of nodes, the reason should be further explained why this method but not other embedding model is selected in the manuscript. Did the authors try other methods for the same aim?

Drug-disease association prediction performance on MSI

Figure R2-1. The drug-disease association prediction performances using four different embedding methods on MSI with random splits. Prediction performance of Heterogeneous SG is significantly higher than that of SG, CBOW and end-to-end neural network embedding methods. CBOW: Continuous Bag-of-words, SG: Skip-gram; t-test rel p: resulting p-value of paired t-test.

We thank the reviewer for the thoughtful comment. The Skip-gram embedding model followed by random-walk based sequence generation is now a standard protocol in graph representation learning. Other widely used alternative algorithms are Continuous Bag-of-words (CBOW) and neural network-based end-to-end embedding methods. In detail, Skip-gram tries to predict the context words given a target word, while CBOW tries to predict the target word given its context words. This enables Skip-gram to handle rarely used words better. CBOW performs well when a target word has many context words, but perform poorly when the target word appears rarely and has few context words. Additionally, Skip-gram can capture the various contexts better for a word with multiple-meanings that can lead to a more nuanced representation.

To address the reviewer's suggestion thoroughly, we used various embedding methods and found that heterogeneous Skip-gram, inspired from Metapath2vec++ (Dong et al., 2013), produces better and effective embedding. We appreciate the reviewer for us to consider word embedding methods more thoroughly. Heterogeneous Skip-gram, which takes node type-specific negative sampling into account, generates better and effective embeddings of drugs and diseases, enabling accurate prediction of drug-disease associations.

The drug-disease association prediction performances of each model on the MSI network are shown in **Figure R2-1**. As illustrated, the heterogeneous Skip-gram, which is aware of node type distribution in the knowledge graph, showed the best performance in accuracy (0.877), area under the receiver-operating curve (AUROC, 0.947) and area under the precision-recall curve (AUPR, 0.944), with statistically significant differences compared to other models as confirmed by paired t-test. We appreciate the reviewer's suggestion, which led to the improvement of our model with the heterogeneous Skip-gram approach. We have included these information in the Methods section and Supplementary Figure 5 of the revised manuscript.

(2) There are some small mistakes. Thus, please make sure that it has been proofread to avoid possible grammar and typology errors. For example, in line 76 'Based on these ideas, we propose DREAMwalk: Drug Repurposing through Exploring Associations using Multi-layer random walk'.

DDA prediction performance on MSI network

Figure R2-2. The drug-disease association prediction performances following the change in similarity cut-off on MSI network. Teleport factor was fixed at 0.3.

Following this comment, we did our best to improve the English quality of the manuscript. In particular, we have corrected the phrase pointed out in line 76 to:

“Based on these ideas, we propose DREAMwalk, which stands for: Drug Repurposing through Exploring Associations using Multi-layer random walk.”

(3) The author omitted some citations in the manuscript.

We are embarrassed, realizing such mistakes. We checked and corrected all citations throughout the manuscript.

(4) The methodology section is a little bit simple.

We appreciate your feedback and have carefully considered your concerns regarding the simplicity of the methodology section. We revised the methodology section to provide a more detailed and comprehensive explanation of our approach, including the specific statistical tests and embedding methods used. Specifically, we added new subsections to describe the distance-based analysis and provided a more detailed description of the heterogeneous Skip-gram model and edge-type transition matrix generation, along with gene set enrichment analysis. Additionally, we provided more information about the drug repurposing candidate selection process, which is reflected in Figure 1 of the revised manuscript. The revised text is marked in blue in the main text.

(5) The authors fused various information, is there any noisy information among these different data?

Thank you for your insightful feedback. We acknowledge your concern about the presence of noisy information in our biomedical knowledge graph. There can be incorrect or noisy information in any databases, including knowledge graphs that we used for this study. In this case, what we can do is to come up with criteria for confidence on similarity measures that we used in this study. For drug-drug similarity, we empirically set up a cutoff value of 0.4 and for disease-disease similarity, we empirically set up a cutoff value of 0.4.

DDA prediction performance of DREAMwalk on MSI network

Figure R2-3. The drug-disease association prediction performances After 10-Fold and 5-Fold cross validation on the MSI network. t-test rel p : resulting p -value of paired t-test.

To support our strategy, we performed experiments with varying cutoff values ranging from 0.1 to 0.9, with the best performance achieved at a cut-off value of 0.4 (Figure R2-2). We infer from this result that similarities below 0.4 can be considered noisy information that decreases DDA prediction performance.

We acknowledge that the presence of noisy information can impact the accuracy of our predictions. We have taken steps to minimize this issue and optimize the performance of our algorithm. Our results demonstrate the effectiveness of our approach in reducing noise and balance the amount of information to be fused during the integration of various information sources, including biomedical knowledge graph and semantic information.

(6) Why was ten fold cross-validation chosen instead of the usual five fold?

We appreciate your question regarding the choice of ten-fold cross-validation in our study. The main reason we selected ten-fold cross-validation (CV) is to obtain more reliable estimates of the model's performance.

It has been demonstrated that increasing the number of folds in cross-validation leads to a reduction in variance in the estimate of model performance, as well as a decrease in bias towards overly optimistic estimates of model performance (Kohavi, 1995). While five-fold cross-validation is a commonly used approach, we opted for ten-fold cross-validation to obtain a more robust estimate of our model's performance. Moreover, ten-fold cross-validation is a well-established method in the academic field, and we believe it provides a more accurate assessment of the performance of our model, along with the comparison models.

Nonetheless, we performed a 10-times 5-fold cross-validation for comparison of our model's performance relative to a 10-times 10-fold cross-validation setting (Figure R2-3). The results show that even though the 10-fold cross-validation scheme resulted in a slightly higher performance (accuracy 0.877, AUROC 0.944, AUPRC 0.947) over 5-fold cross-validation (accuracy 0.875, AUROC 0.944, AUPRC 0.946), the difference was not statistically significant (paired t-test p -value < 0.5). Also, the standard deviation was lower in the 10-CV setting (accuracy 0.0048, AUROC 0.0041, AUPRC 0.0043) than 5-CV (accuracy 0.0054, AUROC 0.0045, AUPRC 0.0049), as expected. We hope that these explanations and results addresses your concerns.

Figure R3-1. The all-pairwise normalized Euclidean distance between treatments with shared indications. Due to the underflow issue of python, any value of below $10E-308$ is represented as 0.0. t-test rel p: resulting p-value of paired t-test.

Reviewer #3 (Remarks to the Author):

This paper describes a new method, DREAMwalk, for performing node embedding on biomedical knowledge types. It adapts existing and established random walk techniques by randomly “teleporting” to nodes of similar types when performing random walks. The authors use their algorithm to propose (1) potential mechanisms of action by which existing drugs operate in the body and (2) potential drug repurposing candidates for Alzheimer’s disease and breast cancer.

VALIDITY

The key conclusions of the paper are as follows:

DREAMwalk embeds drugs with similar use closer together than baselines. This seems to be supported qualitatively by Figure 3 but is not quantitatively substantiated in general.

We thank the reviewer for the valuable comment. We appreciate the concern that even though we have qualitatively shown the cases reflecting the embedding space’s characteristics of locating drugs with similar use closer, we have not been able to show them quantitatively. To address this concern, we have performed an additional experiment to quantitatively assess whether the drugs with similar use are closer, compared to the method without the ‘teleportation’ to semantically close neighbors.

For the experiment, we selected four diseases with the most treatments in the MSI biomedical knowledge graph (biomedKG), namely rheumatoid arthritis, asthma, hypertension, and allergic rhinitis. Each four diseases are associated with drugs of varying mechanism of actions. Then, we calculated the all-pairwise normalized Euclidean distances between the embedding vectors of the drugs, in both teleport-guided embedding space and non-teleported embedding space. Then, we reported the resulting p-value of the paired t-test to validate the statistical significance of the difference.

As a result, drugs of all four diseases were significantly closer when teleport operation was applied (**Figure R3-1**). This result is in line with our prior qualitative results on the previous version of the manuscript, and also provides a more generalized substantiation. We have improved the manuscript by adding the results above to the Results section.

a Gene set enrichment results shown in the previous version of manuscript

b GO enrichment analysis of Gabapentin's window neighbors

c KEGG enrichment analysis of Parkinson's disease's window neighbors

Figure R3-2. The window neighbor gene set analysis. **a** The gene set enrichment analysis result figure from the previous version of the manuscript. The terms were sorted in a DREAMwalk-preferred way. **b,c** The rearranged gene set enrichment analysis results for gabapentin and Parkinson's disease (PD), respectively. The GO terms/KEGG pathways that are not present in the comparison list are starred.

DREAMwalk identifies drug mechanisms of action better than existing methods. This given data seem insufficient to properly support this conclusion. Most of the results related to MoA prediction are summarized in Figure 5. In this figure, mechanism of action is identified by plugging a group of “window neighbor” proteins (nodes within k steps of the drug in a random walk) into a third party grouping tool, which identifies mechanisms and statistical significance. Mechanisms were sorted in descending order relative to paths produced by DREAMwalk. However, the authors fail to show mechanisms sorted in descending order of the baselines, which does not allow for an unbiased comparison. This makes it difficult to determine if DREAMwalk is truly better than baselines.

We appreciate the reviewers' concern regarding the potential for biased results in the arrangement of our figure. We agree that presenting only the enrichment results in descending order of DREAMwalk paths can lead to unfair comparison with baseline methods. To address this issue, we have divided the plots in Figure 5 into two separate lists, one for teleport-guided paths and the other for non-teleported paths, each arranged in descending order of their respective statistical significance (**Figure R3-2**).

Among the exclusive list of terms between teleport-guided and non-teleported paths, our teleport-guided approach identified terms associated with critical mechanism of action, for example GABA receptor activity, GABA-gated chloride ion channel activity, GABA-A receptor activity for gabapentin and Fluid stress and atherosclerosis, Focal adhesion, Apoptosis pathways for Parkinson's disease.

By presenting the results in this way, we not only show that DREAMwalk identifies more relevant terms in terms of Mechanism of Action than the non-teleported path, but also that our approach produces more statistically significant results, as supported by more significant adjusted p-values by Bonferroni multiple test adjustment and higher gene overlap ratios of the enriched terms. Overall, we believe that this new arrangement allows for a more unbiased comparison of the enriched terms and their statistical test results. We thank the reviewers for bringing this issue to our attention and have included the revised figure in the Results section of the manuscript.

DREAMwalk identifies novel & meaningful drug repurposing candidates better than baselines. The given data are not sufficient to support this conclusion. First, Table 1 omits a number of highly ranked drug repurposing candidates. All of the top ranked candidates should be included here because it allows the reader to better evaluate potential failure modes of the algorithm. It would additionally be helpful to compare the top candidates produced by DREAMwalk to those by another algorithm, e.g. residual2vec to see how they compare and if DREAMwalk's candidates are demonstrably better.

We appreciate the reviewers' critical comment regarding the need to provide a comprehensive list of top-ranked drug repurposing candidates to better evaluate the performance of our algorithm. While we initially thought that listing only the drugs well-supported by the literature would be sufficient, we agree that providing the complete top-ranked list and comparing it to those produced by other algorithms would better demonstrate the clinical applicability of our model.

To address this issue, we have included the full top-10 prediction list of our model in **Table R3-1**. It is important to note that we have updated our model from using a Multi-layer perceptron (MLP) classifier to an eXtreme Gradient Boosting (XGBoost) classifier, which resulted in a different drug top-10 list from the previously submitted version. In addition, we have provided the top-10 list generated by baseline models, namely DTi2vec, NEWMIN, WalkPool, SEAL, ComplEx, and edge2vec (**Table R3-2**).

Table R3-1. Top-10 list of the drug repurposing candidates identified by DREAMwalk for Breast carcinoma and Alzheimer's disease. Avg. Prob: average probability; SD : Standard deviation

Breast Carcinoma				
Rank	Drug	Original indication	Avg. Prob.	SD
1	Irinotecan	Colorectal cancer, SCLC, NSCLC	0.9949	0.00332
2	Etoposide	Germ cell tumors, Kaposi sarcoma, SCLC	0.9930	0.00662
3	Dactinomycin	Wilm's tumor, Rhabdomyosarcoma, Neuroblastoma	0.9916	0.01278
4	Teniposide	ALL, Small Cell Carcinoma, Lymphoid Leukemia	0.9906	0.01091
5	Vinblastine	Hodgkin disease, Lymphoma, NHL	0.9890	0.01246
6	Mitoxantrone	AML, Multiple Sclerosis, Lymphoma, Sarcoma	0.9888	0.01279
7	Interferon alfa-2b	Melanoma, Brain Neoplasms, Hepatitis C	0.9884	0.00954
8	Cortisone-acetate	SLE, CTCL, IBD, Autoimmune Diseases	0.9871	0.02182
9	Vindesine	CML, Melanoma, ALL, Hodgkin's lymphoma	0.9870	0.01003
10	Hydroxyurea	CML, cancer of head and neck, sickle cell anemia	0.9862	0.01279
Alzheimer's disease				
Rank	Drug	Original indication	Avg. Prob.	SD
1	Levetiracetam	Simple partial seizures, Epilepsy	0.9901	0.01375
2	clomipramine	OCD, Chronic pain, Narcolepsy	0.9825	0.01614
3	duloxetine	Major depressive disorder, Peripheral neuropathy	0.9767	0.02362
4	fluoxetine	Major depressive disorder, OCD, Bipolar disorder	0.9755	0.0315
5	maprotiline	Depressive disorder, Duodenal Ulcer	0.9743	0.02527
6	armodafinil	Narcolepsy, Obstructive sleep apnea	0.9737	0.02783
7	sertraline	Depressive disorder, OCD, Panic disorders	0.9731	0.03506
8	Lisdexamfetamine	ADHD	0.9711	0.03051
9	atomoxetine	ADHD	0.968	0.05088
10	dextroamphetamine	ADHD, narcolepsy	0.9666	0.03925

Comparing the repurposing lists for breast carcinoma with NEWMIN or edge2vec's list, we observed that while NEWMIN and edge2vec are not successful in finding drug candidates from interferon derivatives and corticosteroids (drugs with suffix '~sone/lone'), DREAMwalk ranked highly not only chemotherapeutic agents but also interferon derivatives and corticosteroids. These results demonstrate the ability of our model to identify novel and meaningful drug repurposing candidates.

We thank the reviewers for highlighting the importance of providing a comprehensive list of top-ranked drug candidates and have included the full top-10 repurposing list of our model and baseline models in the Results section and Supplementary Table 5 of the revised manuscript.

SIGNIFICANCE

The proposed model makes an incremental improvement on existing graph embedding methodology. Most of the baselines compared with are relatively old and outdated -- more recent models such as SEAL show better link prediction performance on public benchmarks. Additionally, this paper does not consider more general knowledge graph link prediction methods (e.g. ComplEx, RotatE, QuatE, etc) to see if this class of model may have better drug-disease association (DDA) performance than methods investigated.

We appreciate the thorough review and thank the reviewer for the feedback. We have updated our manuscript to include additional state-of-the-art models for comparison such as SEAL and general knowledge graph link prediction methods (ComplEx, RotatE, QuatE), as suggested by the reviewer. We also included WalkPool, a state-of-the-art link prediction model on Papers with Code benchmark (<https://paperswithcode.com/task/link-prediction>). Additionally, we introduced two recent models for link prediction on biomedKGs, DTi2vec and NEWMIN. We provide the link prediction results for all our benchmark tasks (**Figure R3-3**) as well as drug repurposing list for breast carcinoma and Alzheimer's disease (**Table R3-2**).

Table R3-2. Top-10 drug repurposing candidates for Alzheimer’s disease and breast carcinoma of baseline models.

Alzheimer’s disease						
Rank	DTI2vec	NEWMIN	WalkPool	SEAL	Complex	edge2vec
1	promazine	clonazepam	isoflurophate	Zinc	dopamine	dexamethasone
2	melatonin	carbamazepine	mestifon	Cyclophosphamide	amantadine	hydrocortisone
3	felbamate	propentofylline	Choline	Ritonavir	Phenobarbital	prednisolone
4	tetrabenazine	sertraline	choline salicylate	Rasagiline	norepinephrine	triamcinolone
5	gabapentin	topiramate	dipivefrine	Melatonin	ifosfamide	Interferon alfa-2b
6	diazepam	tiagabine	Mecasermin	Resveratrol	leuprolide	fludroxycortide
7	lorazepam	levetiracetam	edrophonium	Mecasermin	Theophylline	prednisone
8	phenytoin	ramelteon	pralidoxime	Tranylcypromine	cisplatin	rimexolone
9	clonazepam	retigabine	pralidoxime-chloride	Diacerein	baclofen	methylprednisolone
10	Methamphetamine	Chlordiazepoxide	demecarium	Choline salicylate	orphenadrine	buphenine
Breast Carcinoma						
Rank	DTI2vec	NEWMIN	WalkPool	SEAL	Complex	edge2vec
1	Canakinumab	Interferon alfa-2b	Thalidomide	Succinic acid	etoposide	dexamethasone
2	methotrexate	vinblastine	fostamatinib	Aspirin	vincristine	hydrocortisone
3	Interferon alfa-2b	Peginterferon alfa-2a	dextrazoxane	Arsenic trioxide	hydroxyurea	triamcinolone
4	vinblastine	carmustine	teniposide	dexamethasone	irinotecan	prednisolone
5	Thalidomide	Interferon Alfa-2a	etoposide	arsenic-trioxide	cisplatin	prednisone
6	thiotepa	vincristine	daunorubicin	prednisolone	dactinomycin	Interferon alfa-2b
7	methylprednisolone	Interferon alfacon-1	clofarabine	hydrocortisone	vinblastine	fludroxycortide
8	cisplatin	hydroxyurea	dexamethasone	amcinonide	bleomycin	buphenine
9	Interferon beta-1a	Interferon beta-1a	dactinomycin	urea	carmustine	methylprednisolone
10	azathioprine	Peginterferon alfa-2b	cytarabine	prednisone	ifosfamide	Interferon beta-1b

Drug-disease association prediction performance

Figure R3-3. The drug-disease association prediction performances of each models on the three biomedKGs. **a** DDA prediction performance on MSI network with random split. **b** DDA prediction performance on MSI network with disease split. **c,d** DDA prediction performance with random split on HetioNet and KEGG networks, respectively.

Our results show that DREAMwalk achieves the highest performance over all other methods on drug-disease association (DDA) prediction, even when compared to more recent and advanced models. We observed that the transition-based models, such as ComplEx, RotatE, and QuatE, were not very successful in finding DDAs. We believe that this could be due to their training procedure, which includes learning all association types in the graph, while our experiment settings focused solely on predicting edges between drugs and diseases.

Our additional comparison benchmark with recent link prediction models demonstrates the DREAMwalk's ability to predict drug-disease associations more accurately from heterogeneous biomedKGs, with its generalizability demonstrated from three different graphs. We have included the results of recent model in the Results section of the revised manuscript.

It is worth noting that the vast reported repurposing results in Table 1 seem to identify drugs used for very similar diseases (e.g. other cancer treatments for breast cancer; PD treatments for AD). Without providing any associated context for these predictions, the result indicate that DREAMwalk struggles to identify truly novel drug repurposing candidates beyond what could be deduced via visual inspection by a clinician.

Thank you for the thoughtful comment. We appreciate the opportunity to provide further context and clarification on the drug repurposing candidates identified by DREAMwalk. We acknowledge that the identification of drugs used for similar diseases may not necessarily be considered novel. However, we would like to emphasize that DREAMwalk's ability to identify multiple drugs for the same disease could still be useful in a clinical setting. It can provide additional treatment options if the first-line drugs are not effective.

In the previous version of our manuscript, we only listed the top-10 drugs with supporting literature, which may have given the impression that the model struggled to identify truly novel repurposing candidates. To address this concern, we have included the full top-10 candidate drugs for breast carcinoma and Alzheimer's disease in **Table R3-1**, as mentioned in the previous response in the "Validity" section.

The list includes a variety of drug categories, including chemotherapeutics, interferon, and corticosteroids for breast cancer, and anti-depressants, anti-ADHD, anti-narcoleptic drugs, and anti-epileptic drugs for Alzheimer's disease. While most of the listed drugs are supported by the literature, some are old and some are relatively recent. Additionally, the list includes drugs that are not supported by previous studies, which may be seen as truly novel repurposing candidates. We hope that this additional information addresses your concerns and underscores the potential clinical utility of our model in identifying drugs for repurposing.

DATA AND METHODOLOGY

Performance of MoA prediction needs more data to properly validate (see second bullet of "Validity" section). Novelty of drug repurposing candidates needs more evidence to conclusively demonstrate if DREAMwalk is better than existing methods (see "Validity"). Drug repurposing candidates are omitted from Table 1 making it impossible to determine if best candidates were "cherry picked" or if they are truly representative of all results.

We thank the reviewer for the insightful comments.

We have carefully considered your concerns regarding the validity of our MoA prediction and the novelty of our drug repurposing list. We have addressed these concerns in detail in the relevant sections in the "Validity" section and **Figure R3-2**.

Regarding the drug repurposing candidates listed in Table 1, we apologize for omitting some of the candidates from the table. We have now included the full list of top candidates for each disease in **Table R3-1**. We hope this additional information will help address your concerns and demonstrate that our results are not cherry-picked, but rather representative of the output generated by our model.

We hope that these revisions have adequately addressed your concerns and have helped to clarify any issues in our paper.

ANALYTICAL APPROACH

Paper needs better details of what statistical tests were used to compare score distributions and determine significance.

Thank you for your critical and important feedback. We agree that more details on the statistical tests that were used to compare score distributions and determine significance need to be provided to fully understand the effectiveness of our approach. We have revised the manuscript to clarify the methods we have used in our study.

We have added a new subsection, "Statistical analyses," to the Methods section of the manuscript. This subsection explains in detail the statistical tests used in our study, including the Fischer's exact test comparing the gene sets and the Benjamini-Hochberg correction for multiple test correction. Additionally, we have annotated the statistical tests and their precise results in Figure 3. We believe that these additions will address your concerns and improve the clarity of our methodology. The revised parts are marked in blue in the main text.

Suggested Improvements include:

- Show mechanisms of action and drug repurposing candidates identified by baseline algorithms to show if/where DREAMwalk may be worse than existing methods
- Show all top-ranked candidates
- Use more (and more recent) baselines to compare DDA prediction performance (e.g. KG embedding methods, top methods from link prediction section of Open Graph Benchmark)

We thank the reviewer for the valuable suggestions. We believe the suggestions were all mentioned in the above sections, and have provided the results accordingly. The results on mechanism of action are shown in **Figure R3-2**, while the drug repurposing candidates are presented in **Table R3-1** and **Table R3-2**. Moreover, the performances of the recent baselines are illustrated in **Figure R3-3**.

Regarding the first suggestion, we acknowledge the importance of comparing DREAMwalk with other baseline methods for identifying mechanisms of action and drug repurposing candidates. Our method generates node sequences that include genes, which can be used for explaining MoA. However, it is challenging to identify MoA for drug-disease associations predicted by existing tools since these tools do not produce sets of genes while predicting drug-disease associations.

However, we acknowledge the need for benchmarking and have compared our teleport-guided path generation method with non-teleported path generation to demonstrate the effectiveness of our approach in associating to more plausible biological terms. We believe this comparison validates the use of teleportation in our method and justifies its utility for identifying novel mechanistic insights and drug repurposing candidates.

We would be happy to explore alternative ways of benchmarking our method against other baseline algorithms in future studies.

We thank the reviewer for providing an opportunity to improve the manuscript.

CLARITY AND CONTEXT

This paper needs to include additional graph mining algorithms as performance baselines in identifying drug repurposing candidates. This would give better context and clarity for if and how exactly DREAMwalk's results differ from existing results.

Thank you for the critical and important comment. We used the graph mining methods such as transition-based models ComplEx, RotatE and QuatE, as well as the state-of-the-art link prediction GNN models SEAL and WalkPool, in our comparison analysis, and measured the prediction performances on drug and disease associations. The performances and drug repurposing lists for these models are presented in **Figure R3-3** and **Table R3-2**, respectively, with detailed explanations in the response to the first bullet of the Significance section. We have also added this information to the Results section of the revised manuscript. We hope that the addition of these models provides more comprehensive and insightful comparisons of DREAMwalk's results with existing methods.

References

Thafar, Maha A., et al. "DTi2Vec: Drug–target interaction prediction using network embedding and ensemble learning." *Journal of cheminformatics* 13.1 (2021): 1-18.

Yu, Lian, Mingfei Xia, and Qi An. "A network embedding framework based on integrating multiplex network for drug combination prediction." *Briefings in bioinformatics* 23.1 (2022): bbab364.

Dong, Yuxiao, Nitesh V. Chawla, and Ananthram Swami. "metapath2vec: Scalable representation learning for heterogeneous networks." *Proceedings of the 23rd ACM SIGKDD international conference on knowledge discovery and data mining*. 2017.

Chen, Tianqi, and Carlos Guestrin. "Xgboost: A scalable tree boosting system." *Proceedings of the 22nd acm sigkdd international conference on knowledge discovery and data mining*. 2016.

Jourdan, Jean-Pierre, et al. "Drug repositioning: a brief overview." *Journal of Pharmacy and Pharmacology* 72.9 (2020): 1145-1151.

Cummings, Jeffrey, et al. "Alzheimer's disease drug development pipeline: 2022." *Alzheimer's & Dementia: Translational Research & Clinical Interventions* 8.1 (2022): e12295.

Kohavi, Ron. "A study of cross-validation and bootstrap for accuracy estimation and model selection." *Proceedings of the 14th international joint conference on Artificial intelligence-Volume 2*. 1995.

Reviewer #1 (Remarks to the Author):

I thank the authors for thoroughly revising this manuscript by performing additional statistical analyses, comparing the new approach to existing drug repurposing techniques and knowledge graph embeddings, and revising the text to include a discussion on model performances and findings. I have no additional requests.

Reviewer #2 (Remarks to the Author):

I confirm that all my comments on the previous version of the paper have been addressed satisfactorily. I recommend that this paper can be accepted without any revisions.

Reviewer #3 (Remarks to the Author):

The authors have substantially improved the quality, transparency, reproducibility, and utility of their manuscript and algorithm. The addition of full candidate list results with statistical significance, teleport results for mechanism of action, and a more complete list of recent baselines of other knowledge graph model comparisons have fixed the primary technical weaknesses identified in the initial submission. While the underlying technology is not entirely novel compared to the scope of other recent state-of-the-art knowledge graph works, it has been well vetted and provides a comprehensive assessment of its ability in the drug repurposing space.